# Clinical Features, Gene Alterations, and Outcomes in Prefibrotic and Overt Primary and Secondary Myelofibrotic Patients

**DOI:** 10.3390/cancers14184485

**Published:** 2022-09-16

**Authors:** Tong-Yoon Kim, Daehun Kwag, Jong-Hyuk Lee, Joonyeop Lee, Gi-June Min, Sung-Soo Park, Silvia Park, Young-Woo Jeon, Jae-Ho Yoon, Seung-Hawn Shin, Seung-Ah Yahng, Byung-Sik Cho, Ki-Seong Eom, Yoo-Jin Kim, Seok Lee, Hee-Je Kim, Chang-Ki Min, Seok-Goo Cho, Jong-Wook Lee, Jong-Mi Lee, Myungshin Kim, Sung-Eun Lee

**Affiliations:** 1Department of Hematology, Seoul St. Mary’s Hospital, College of Medicine, The Catholic University of Korea, Seoul 06591, Korea; 2Department of Hematology, Yeouido St. Mary’s Hospital, College of Medicine, The Catholic University of Korea, Seoul 06591, Korea; 3Leukemia Research Institute, College of Medicine, The Catholic University of Korea, Seoul 06591, Korea; 4Department of Hematology, Eunpyeong St. Mary’s Hospital, College of Medicine, The Catholic University of Korea, Seoul 06591, Korea; 5Department of Hematology, Incheon St. Mary’s Hospital, College of Medicine, The Catholic University of Korea, Seoul 06591, Korea; 6Department of Laboratory Medicine, Seoul St. Mary’s Hospital, College of Medicine, The Catholic University of Korea, Seoul 06591, Korea

**Keywords:** genetic alteration, prefibrotic primary myelofibrosis, overt primary myelofibrosis, secondary myelofibrosis

## Abstract

**Simple Summary:**

The final stages of myeloproliferative neoplasms (MPNs) are related to myelofibrosis (MF). This study aimed to compare secondary myelofibrosis (SMF), overt (PMF), and prefibrotic primary myelofibrosis (pre-PMF) and determine prognostic factors in clinical and genetic features. This study included 229 patients; 67 (29%), 122 (53%), and 40 (18%) cases were confirmed as SMF, overt PMF, and pre-PMF, respectively. We compare MFs, including pre-PMF, using six scoring stratifications associated with PMF and one scoring system with SMF. We also evaluated the impact of the genetic groups with clustering methods and previously reported genomic groups. We determined the clinical and genetic features associated with disease progression in SMF, overt PMF, and pre-PMF groups.

**Abstract:**

The Philadelphia-negative myeloproliferative neoplasms (MPNs) are divided in three major groups: polycythemia vera (PV), essential thrombocythemia (ET), and primary myelofibrosis (PMF). The 2016 WHO classification incorporates also prefibrotic PMF (pre-PMF) and overt PMF. This study aimed to discriminate the clinical features, genetic alterations, and outcomes in patients with prefibrotic, overt PMF, and secondary MF (SMF). This study included 229 patients with diagnosed myelofibrosis (MF). Among 229 patients, 67 (29%), 122 (53%), and 40 (18%) were confirmed as SMF, overt PMF, and pre-PMF, respectively. The *JAK2* V617F mutation was differentially distributed in SMF and PMF, contradictory to *CALR* and *MPL* mutations. Regarding nondriver mutations, the occurrence of *ASXL1* mutations differed between PMF and SMF or pre-PMF. The three-year overall survival was 91.5%, 85.3%, and 94.8% in SMF, overt PMF, and pre-PMF groups. Various scoring systems could discriminate the overall survival in PMF but not in SMF and pre-PMF. Still, clinical features including anemia and thrombocytopenia were poor prognostic factors throughout the myelofibrosis, whereas mutations contributed differently. Molecular grouping by wild-type *SF3B1* and *SRSF2/RUNX1/U2AF1/ASXL1/TP53* mutations showed inferior progression-free survival (PFS) in PMF, SMF, and pre-PMF. We determined the clinical and genetic features related to poor prognosis in myelofibrosis.

## 1. Introduction

Philadelphia-negative myeloproliferative neoplasms (MPNs) are clonal hematopoietic disorders characterized by the overproduction of differentiated hematopoietic cells, and are divided into three major subcategories, namely, polycythemia vera (PV), essential thrombocythemia (ET), and primary myelofibrosis (PMF). PV and ET can evolve to secondary forms of myelofibrosis (MF), known as post-polycythemia vera (PPV-MF) and post-essential thrombocythemia myelofibrosis (PET-MF).

The progression of myelofibrosis might lead to various pathological conditions, including thrombosis, infection, leukemic transformation, and eventually death [1]. However, the time to progression and median survival time vary between patients with MPN; thus, predicting prognosis or detecting the high-risk group for progression is essential [2]. The 2016 WHO classification has specified prefibrotic PMF (pre-PMF) and overt PMF as indicators for the early detection before profound progression in patients with PMF. A previous study has reported that pre-PMF accounted for approximately 10% of MPN cases, with a survival outcome inferior to ET or PV [3].

Excessive myeloproliferation in Philadelphia-negative MPNs are driven by mutations in *JAK2*, *CALR*, *MPL*, and uncommon variants. Recently, evidence for the acquisition of somatic mutations and information on factors influencing clonal outgrowth such as aging and the bone marrow microenvironment have been accumulating [4]. Approximately one third of patients with Philadelphia-negative MPN have additional mutations that alter DNA methylation (*DNMT3A*, *TET2*, and *IDH1/2*), chromatin modifications (*ASXL1*, *EZH2*, and *IDH1/2*), messenger RNA splicing (*U2AF1*, *SF3B1*, *SRSF2*, and *ZRSR2*), and DNA repair (*TP53*) [5]. Although there was a study on early MF [6], comparison studies between pre-PMF, overt PMF, and SMF data were limited. Patients with PMF possessing the *CALR* type 1/type 1-like mutation have a better prognosis than those with *CALR* type 2/type 2-like, *JAK2*V617F, or *MPL* W515 mutations [7,8,9], and triple-negative cancer [10]. Furthermore, a category of “high molecular risk” (HMR) patients has been defined in patients with PMF who harbor any mutation in *ASXL1*, *EZH2*, *SRSF2*, *IDH1*, and *IDH2* [11,12], and *U2AF1* Q157 [13]. As the studies of mutation order and acquisition accumulates, more interest is in correlation with the clinical course of myelofibrosis and genomic status. Due to clonal evolution during disease progression and various latencies among the genes, order of mutations was different among the patients. Furthermore, mutation order affects the presentation of clinical course and drug susceptibility [14,15,16]. However, knowledge about the molecular profile in secondary MF and current information about the differences between pre PMF and overt PMF are scarce. Here, we evaluated the clinical features, genetic alterations, and outcomes in patients with prefibrotic and overt PMF and SMF to assess the importance of distinguishing them in clinical practice.

## 2. Materials and Methods

### 2.1. Patient Selection 

We included 229 patients who were diagnosed with myelofibrosis at Seoul St. Mary’s Hematology Hospital from December 2001 to August 2021 and provided DNA samples for next-generation sequencing (NGS) analysis. Pathologists confirmed the diagnosis based on the 2016 WHO classification [17].

Most DNA samples were from bone marrow (BM) aspirate samples at the diagnosis of myelofibrosis, regardless of pre-evolving myeloproliferative neoplasm (MPN) disease. This study was approved by the Institutional Review Board of Seoul St. Mary’s Hospital, The Catholic University of Korea, Seoul, Korea (KC22RISI0120) and was conducted in accordance with the tenets of the Declaration of Helsinki.

### 2.2. Molecular and Cytogenetic Studies

Conventional BM karyotyping was performed on G-banded metaphase chromosomes using routine techniques. Karyotypes were interpreted according to ISCN 2016 [18]. 

Regarding molecular analysis, NGS, was performed using a customized myeloid panel (“SM panel”), as described in our previous report [19]. The SM panel contains 87 genes frequently found mutated in patients with MPN (Appendix A). According to the manufacturer’s instructions, target-capturing sequencing was achieved using a customized target kit (3039061; Agilent Technologies (Santa Clara, CA, USA). DNA libraries were composed according to the protocol of the manufacturer, and sequencing was performed using the Illumina HiSeq4000 platform (San Diego, CA, USA). Variants with more than 20 reads and 5% variant allele frequencies (VAFs) were considered to be mutated. Mutations in *JAK2*, *CALR*, and *MPL* of less than 5% VAF were considered positive with a low allele burden. Using the Integrative Genomic Viewer, the detected variants were manually verified.

### 2.3. Definitions

Overall survival (OS) and progression-free survival (PFS) were calculated from the date of diagnosis. OS was defined as the time to death from any cause. PFS was defined as the time to progression to overt myelofibrosis or death in the pre-PMF groups; otherwise, progression to acute leukemia or death was considered the endpoint event.

The scoring system consisted of IPSS [1], DIPSS [20], DIPSS-plus [21], MIPSS70 [22], GIPSS [23], MIPSS70 + Ver2 [13], and MYSEC-PM [24], which were calculated as previously reported. 

### 2.4. Statistical Analyses 

Baseline: The clinical and molecular characteristics of patients classified into the three groups were compared: SMF, PMF, and pre-PMF. The chi-square or Fisher’s exact test were used for categorical variables, whereas the two-sample *t*-test or Mann–Whitney U test were used for continuous variables. 

We used the decision tree method (R packages *rpart* 4.1-10) to discriminate between the MF groups using clinical and genetic variables. To emphasize the effect of the non-driver mutations, we included the other mutations except the *JAK2*, *MPL*, and *CALR* mutations in the clustering analysis. Binary distances were calculated and hierarchical clustering was performed using the Ward’s method. The best proposal cluster-number was calculated using Silhouette, Ratkowsky and Lance, and McClain and Rao indices (R packages *Nbclust* 3.0). 

The OS and PFS were estimated using the Kaplan–Meier method, and groups were compared using the log-rank test. The Cox proportional hazards model was used for univariate and multivariate analyses of OS and PFS. Variables with *p* < 0.10 determined using univariate analysis were considered for multivariate analysis. Statistical analyses were performed using R statistical software (version 3.4.3; R Foundation for Statistical Computing, Vienna, Austria). 

## 3. Results

### 3.1. Clinical and Cytogenetic Features of Patients with Prefibrotic PMF, Overt PMF, and Secondary MF

The clinical and cytogenetic characteristics of patients are summarized in Table 1. Among the patients, PMF was the most common (*n* = 122, 53.3%), followed by PET-MF (*n =* 46, 20.1%), pre-PMF (*n* = 40, 17.4%), and PPV-MF (*n* = 21, 9.2%). In particular, we found that pre-PMF cases had a lower median age (*p* = 0.003), higher hemoglobin levels (*p <* 0.001), higher platelet count (*p* < 0.001), lower peripheral blood blast proportion (*p* < 0.001), smaller spleen (*p* < 0.001), and fewer constitutional symptoms (*p* < 0.001) than other overt MF cases. 

Based on the DIPSS scoring system, we did not identify any differences in unfavorable karyotype abnormalities among the three groups. However, we found that the MIPSS karyotype risk was higher in the PMF group than in the SMF and pre-PMF groups. In the pre-PMF group we did not identify any karyotypes including abnormalities of −7, i(17q), inv(3)/3q21, 12p-/12p11.2, 11q-/11q23, or any other autosomal trisomy not including +8, +9 and complex karyotype.

### 3.2. Genetic Features of Patients with Prefibrotic PMF, Overt PMF, and Secondary MF

We identified mutations in 92.1% of patients (*n* = 211). The mean number of mutations was 1.80 (range 0–6) in total, including 1.78 (range 1–5) in SMF, 1.91 (range 0–5) in PMF, and 1.65 (range 0–6) in pre-PMF. The genetic landscape in the three groups of MF is shown in Figure 1A, and a comparison of proportion among PMF, SMF, and pre-PMF is presented in Figure 1B. We found that in patients with PMF, mutations in *JAK2* were the most common (*n* = 57, 46.7%), followed by those in *ASXL1* (*n* = 43, 35.2%), *CALR* (*n* = 34, 27.9%), *TET2* (*n* = 11, 9.0%), and *U2AF1* (*n* = 8, 6.6%). Similarly, in the SMF group, mutations in *JAK2* were the most common (*n* = 43, 64.2%), followed by those in *ASXL1* (28.4%), *CALR* (26.9%), *TET2* (13.4%), and *SF3B1* (7.5%). In the pre-PMF group, mutations in *JAK2* were the most common (50.0%), followed by those in *CALR* (17.5%), *TET2* (17.5%), *SF3B1* (10.0%), and *DNMT3A* (10.0%). However, it was found that the proportion of *ASXL1* mutations in the pre-PMF group was lower than that in the SMF and PMF groups (Table 1).

The median variant allele frequency (VAF) of gene mutations was 0.37 overall, 0.36 in PMF, 0.39 in SMF, and 0.38 in pre-PMF. In PMF, SMF, and pre-PMF, 8.2%, 16.5%, and 10.5% of homozygous mutations had a higher VAF of 0.70. Except for JAK2 (0.54 vs. 0.62 vs. 0.34, *p* = 0.001), no differences were detected in VAFs of the examined genes, including *CALR*, *MPL*, *ASXL1*, *TET2*, *SF3B1*, *DNMT3A*, *MPL*, *U2AF1*, and *RUNX1* (Figure 1C). Next, we analyzed the correlation of gene mutations detected in more than three patients (Appendix A), and investigated co-occurrence driver mutations in *JAK2*, *CALR*, and *MPL* with other mutations. In the PMF group, mutated *JAK2* and *TP53* showed negative correlations (−0.19, *p* = 0.033). Additionally, *CALR* and *TET2* were positively correlated (0.31, *p* < 0.001), In the pre-PMF group, we identified a negative correlation between mutated JAK2 and *TET2* (−0.33, *p* = 0.038). 

### 3.3. Correlation of Genetic and Clinical Features 

We further analyzed the correlation of these gene mutations with clinical characteristics. In the PMF group, we found that the *U2AF1* mutation was associated with thrombocytopenia. In the pre-PMF group, we noticed that higher leukocyte counts were negatively correlated with *IDH2* and *ASXL1* mutations. The peripheral blast was negatively correlated with the *ASXL1* mutations. Anemia was positively correlated with the *JAK2* and *DNMT3A* mutations. Thrombocytopenia was positively correlated with *U2AF1* and *MPL*, whereas thrombocytopenia was negatively correlated with the *TET2* mutations (Figure 1D).

To identify the factors in order of importance and to discriminate MF groups, we performed the decision-tree method using clinical and genetic variables. A representative clinical decision tree is shown in Figure 1E. According to the decision tree, variables such as thrombocytopenia, anemia, and peripheral blast counts were more dominant in the SMF and PMF groups than in the pre-PMF group. The distribution of the MF groups is shown relative to hemoglobin counts (*X*-axis) and platelet counts (*Y*-axis) (Appendix A). In addition, the SMF group was primarily associated with older age without the *ASXL1* mutation. Among 65 patients that who did not have thrombocytopenia, anemia, peripheral blasts, and *ASXL1* mutations, most of the patients were diagnosed with pre-PMF (30/65, 46.1%) (Figure 1E). 

### 3.4. Distribution of Risk Categories, Outcomes, and Prognostic Effect of Risk Stratification Systems in Each Subgroup

The distribution of the risk stratification system in each subgroup is described in Table 2. We did not find statistical differences in the scores in IPSS, DIPSS, DIPSS-plus, MYSEC-PM, and MIPSS70 + Ver2 between SMF and PMF. Instead, these risk groups showed differences between the pre-PMF group and the other overt MF groups. Despite differences in the GIPSS score between the pre-PMF and PMF groups, no differences were observed between the SMF and pre-PMF groups. The effect of risk stratification systems on OS and PFS in each subgroup is described in Table 3. All scoring systems could discriminate overall survival in PMF, but not in the SMF and pre-PMF groups. In terms of PFS, we found that scores could predict the progression of pre-PMF to either overt-PMF or secondary acute myeloid leukemia (AML) in IPSS, DIPSS DIPSS-plus, MIPSS70, and MYSECPM, but not in GIPSS. These results implied that prognostic differences between pre-PMF and other MF depended more on the effect of clinical variables than that of genetic variables. The survival outcomes did not change when patients who underwent transplantation were censored at the time of their transplant. 

### 3.5. Univariate and Multivariate Analyses for OS and PFS in Each Subgroup

After a median follow-up of 2.7 years (range, 0.2 to 19.9 years), we observed that the 3-year overall survival of patients was 85.3%, 91.5%, and 94.8% in the overt PMF, SMF, and pre-PMF groups (*p* = 0.026). We recorded 30 deaths among all patients (22 within PMF, 6 within SMF, and 2 within pre-PMF). Among all patients, 48 patients underwent allogeneic hematopoietic stem cell transplantation, including 31 in PMF, 15 in SMF, and 2 in pre-PMF. Among the patients who underwent transplantation, we recorded 10 deaths in the PMF group and 3 deaths in the SMF group.

The results of the univariate analysis are illustrated in Figure 2. In the pre-PMF group, we found that thrombocytopenia, *ASXL1*, *MPL*, *U2AF1*, *SETBP1*, and *SRSF2* mutations were associated with inferior overall survival. In contrast, female sex, constitutional symptoms, anemia, peripheral blood blasts, and *ASXL1* and *SRSF2* mutations were correlated with inferior PFS. We then performed multivariate analysis using the clinical and genetic variables especially for the patients with pre-PMF (Table 4). In model I, we observed that *ASXL1* and anemia were associated with poorer outcomes in PFS. In contrast, in model II, *SRSF2* mutations were correlated with inferior PFS. These models highlighted the effect of genetic mutations on outcomes, such as inferior survival, leukemic transformation, and fibrotic progression. In patients with PMF, the multivariate analysis showed that the *TP53* mutation (*p* = 0.020, HR = 9.29, [95% CI 1.42–60.7]) was associated with poor OS, whereas *CARL* type 1 mutation (*p* = 0.056, HR = 0.12, [95% CI 0.01–1.05]) correlated with favorable PFS. 

In patients with SMF, the multivariate analysis showed that *TP53* and female sex were associated with inferior OS. We also found that female sex, thrombocytopenia, and *RUNX1*, *TP53*, *ZRSR2*, and *IDH1* mutations were associated with inferior outcome in PFS.

### 3.6. Genomic Subgroups in Myelofibrosis by Nondriver Mutations

To analyze the effect of nondriver mutations in MF, we performed hierarchical clustering of the 229 patients according to somatic mutations except the *JAK2*, *MPL*, and *CARL* driver mutations by five subgroups to calculate the specified mentioned index. Grouping nomenclature indicated the most distributed gene mutations in each group (Appendix A). We applied the clustering group to the estimation of overall survival. We accordingly found that 3-year OS was 93.0% 72.9%, 87.6%, 100%, and 100% in clusters 1–5, respectively (Figure 3A,B). Interestingly, the group dominated by *U2AF1*/*TP53* showed the worst PFS in patients with the PMF and SMF groups compared with that in other groups (*p* = 0.258, *p* = 0.023) (Figure 3C,D). However, in the pre-PMF group, the *SRSF2*/*ASXL1* dominant group showed the worst PFS (Figure 3E). This result indicated that each genetic clustering subgroup differently contributed to survival outcomes in each histologically distinct group. However, as *TP53* mutations were not found in pre-PMF, the interpretation of these results requires caution.

### 3.7. Proposal of High-Risk Mutation Groups Predicting Survival Outcomes

First, we evaluated the effect of the genetic mutations, the ARCH (age related clonal hematopoiesis)/CHIP (Clonal Hematopoiesis of Indeterminate Potential) for fibrotic progression, which were previously reported by Bartels, et al. [25]. Among the ARCH/CHIP-associated mutations, the group of mutations was associated with later progression. that is, mutations in *SRSF2, U2AF1, SF3B1, IDH1/2*, and *EZH2*. These mutations were associated with poor OS in the pre-PMF group (*p* = 0.008) (Appendix A) but not in the SMF and PMF groups in our study. 

Next, we analyzed the effect of high-risk genetic mutations for the progression of myelodysplastic syndrome to AML [26], that is, wild type *SF3B1* or mutations in *SRSF2*, *RUNX1*, *U2AF1*, *ASXL1*, and *TP53*, on the survival outcomes in the MF cohort (Figure 4). We observed that the high-risk mutation group discriminated the 3-year OS among all MF subtypes, including the PMF (90.8% vs. 51.1%, *p* < 0.001), SMF (92.2% vs. 83.3%, *p* = 0.009), and pre-PMF groups (100% vs. 50% *p* < 0.001). We also identified that this high-risk subgroup showed inferior PFS in the PMF, SMF, and pre-PMF groups (*p* < 0.001, *p* = 0.024, *p* = 0.026).

## 4. Discussion

Although several previous studies have compared pre-PMF versus ET [27], pre-PMF versus PMF [6,28], and PMF versus SMF [29], to date, no other study except the present study has compared the clinical and genetic characteristics of patients with pre-PMF versus overt PMF and secondary MF. 

A comparison of the clinical features between pre-PMF and PMF showed that patients with pre-PMF were characterized by higher hemoglobin levels and platelet counts, whereas they were less frequently diagnosed with increased peripheral blood blasts, symptoms, and extensive splenomegaly. These findings are consistent with those of a previous study [6]. Similarly, with the exception of old age, a comparison of the clinical characteristics between pre-PMF versus SMF showed a similar trend between the groups. In clinical practice, pre-PMF is typically diagnosed when leukocytosis or thrombocytosis is detected, which is predicted to progress to myelofibrosis in the case of occurrence of cytopenic features such as anemia, thrombocytopenia, or other clinical symptoms during follow-up. 

Comparison of the factors related to survival outcomes obtained in our study with those obtained from the literature revealed certain differences. Among the previously reported variables affecting OS in PET-MF [29] and SMF [24], thrombocytopenia was the only contributing factor to poorer OS in our cohort. Unlike the finding of Makarova et al., that is, 11% of patients with SMF underwent a leukemic transformation, we observed only 2.2% leukemic transformation. These discordances might have occurred because the previous studies were performed in the 1980s, whereas in the present study, MF was first diagnosed in 2001. MYSEC-PM showed a trend toward inferior PFS in intermediate-2 and high groups than the favorable and intermediate-1 group (*p* = 0.059), which is consistent with the results of previous studies. 

We analyzed the effect of individual genes on MF by comparing different groups. In this study, *ASXL1* mutations were relatively infrequent in the pre-PMF group compared with those in the overt MF group. This result is consistent with the previous reports that the presence of *ASXL1* mutations also predicted progression [30,31] in the pre-PMF group. Furthermore, we found that the mean VAF in *JAK2* was low in pre-PMF relative to that in overt MF. Based on these findings, homozygosity in the *JAK2* mutations or higher VAF indicates higher vulnerability to progression, as shown in ET or PV cases [32]. 

Courtier et al. suggested the existence of different mutational landscapes in patients with SMF and PMF, owing to the order of occurrence of mutations. They suggested that RNA splicing genes such as *SRSF2* and *U2AF1* played a role in PMF, which had a higher incidence than that of SMF [33]. Among the RNA splicing genes, *U2AF1* is vital for hematopoiesis and is associated with thrombocytopenia in PMF [34,35]. In the present study, the overt PMF and pre-PMF groups showed a positive correlation with *U2AF1* mutations and thrombocytopenia; however, this correlation was not observed in the SMF group. *SRSF2* mutations were associated with progression in pre-PMF and showed a trend in PMF; however, they were not associated with progression in SMF. However, *ZRSR2* mutations were associated with progression in SMF; this mutation has a higher incidence in SMF than in PMF [33].

Our findings on the progression-related genes in PMF showed some discrepancies with those reported by Vannucchi et al. In their study, *ASXL1*, *EZH2*, *SRSF2*, or *IDH1*/*2* were defined as molecular factors of high risk in PMF [11]. However, these high-molecular-risk-group genes were not associated with significant differences in survival. Instead, mutations in *TP53* and *U2AF1* were associated with poor OS. In addition, mutations in *RUNX1*, *TP53*, and *U2AF1* were also correlated with inferior PFS. However, Grinfeld et al. reported that leukemic transformation is related to the occurrence of mutations in *RUNX1*, *TP53*, and *U2AF1* in MPN disease, which are consistent with our results [36]. These differences might be caused by the different characteristics of the study cohort and the treatment strategies that we are actively implementing at HSCT for transplant candidates. The findings indicated that HSCT reduced progression-related death and that leukemic transformation was a notable contributing factor to survival at our center. 

Patients had an average of 1.8 mutations, making it difficult to use a single gene as a prognostic factor. This finding was due to varying gene frequencies. The single effect of univariate analysis did not reflect the co-occurrence effect. We compensated this limitation by using a hierarchical clustering method of non-driver mutations. Each subgroup was further divided into sets based on the dominance of either chromatin modifications (*ASXL1*) or RNA splicing genes (*U2AF1*, *SRSF2*, and *SF3B1*). Interestingly, *ASXL1, EZH2, SRSF2, IDH1*, and *IDH* mutations were dominant in the third cluster, which was known to be a high-risk group in PMF. This finding indicated that mutations in different genes that frequently occur together need to be considered in addition to the single-gene effect. Within the SMF and pre-PMF groups, the second and third clusters, respectively, showed poor PFS. These findings implied that single genes and clusters of non-driver mutations affect MF survival differently among each group. It might be due to different genetic evolution timing and proportions in each histological MF subtype.

To identify the most influential mutation group in PMF in the literature, we applied the concept suggested by Courtier, et al. [33]. They suggested that mutations that could lead to CHIP and asymptomatic phase of MDS could have an impact different from that of other driver mutations. Therefore, to prove the prognostic effect of previously reported ARCH/CHIP for rapid fibrotic progression in pre-PMF [25] and for leukemic transformation in MDS [26] in the MF cohort, we analyzed an association between these mutations and outcomes (OS and PFS) in patients with prefibrotic, overt PMF, and SMF. Mutations pertaining to fibrotic progression [25] did not contribute to the differences in survival except in pre-PMF, whereas leukemic transformation high-risk mutations [26] showed differences in survival outcomes among all MF subgroups. These findings show that failure in bone marrow functioning contributes to the occurrence of MF.

Our study had several limitations. First, owing to the relatively small number of patients with pre-PMF, it is difficult to draw definite conclusions. Second, our study cohort was limited to a single center; thus, further studies are needed to verify the applicability of our findings to other groups. However, our study demonstrated the clinical and genetic characteristics of MF subtypes, which is important for discriminating secondary MF and prefibrotic MF. In this study, we showed that current risk stratifications are still important for predicting the outcome in patients with overt PMF. However, a widely known GIPSS in overt PMF could not correctly predict the outcome in SMF and pre-PMF. 

## 5. Conclusions

In this study, we investigated distinct clinical and genetic characteristics of patients with prefibrotic PMF, overt PMF, and SMF. Our findings further suggested the mutation profiles that could be widely applied for the stratification of patients with MF, including those with pre-PMF. Using this molecular profiling, symptomatic patients with pre-PMF could be candidates for close monitoring and/or therapeutic application such as *JAK2* inhibitor therapy [37]. Furthermore, our findings raise the possibility that inhibitors targeting *SRSF2* mutations [38] or BET inhibitors [39] could be attempted in the high-risk MF group. Additional large cohort studies with a longer follow-up are needed to confirm the accurate disease-associated molecular characteristics in patients with prefibrotic, overt PMF, and SMF.

## Figures and Tables

**Figure 1 cancers-14-04485-f001:**
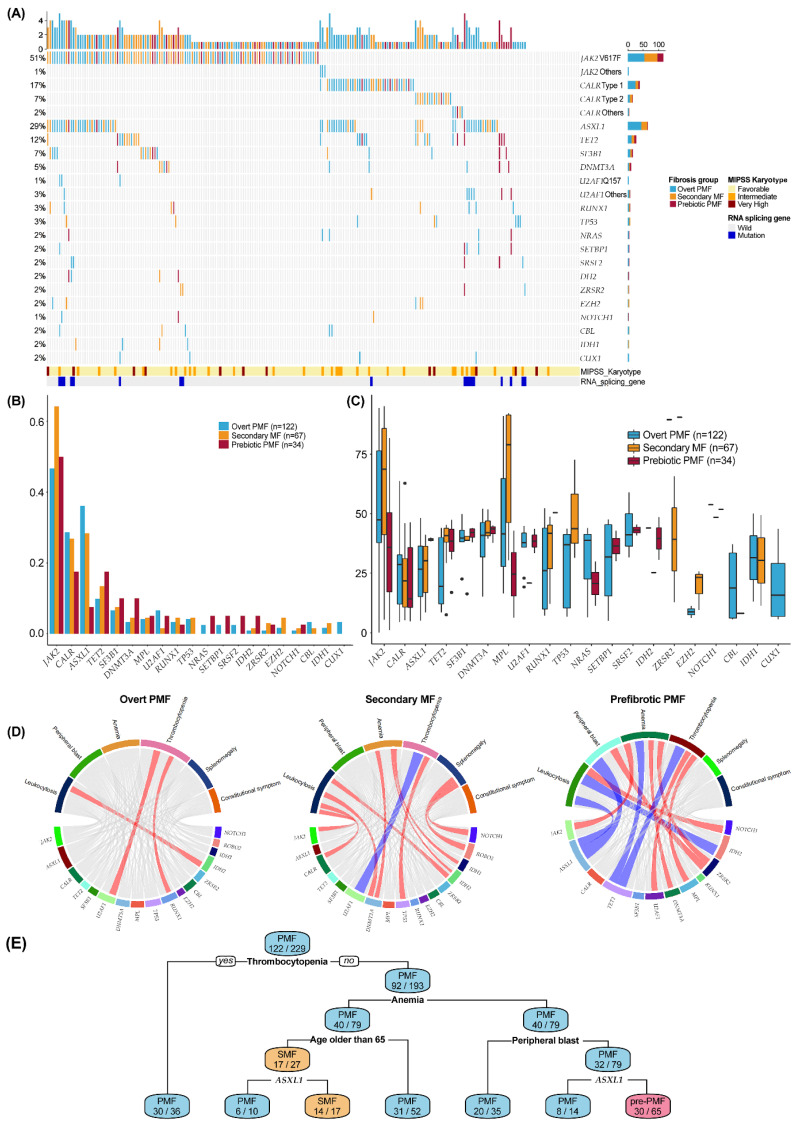
Mutational spectrum in overt primary myelofibrosis (PMF), secondary myelofibrosis (SMF), and prefibrotic myelofibrosis (pre-PMF). (**A**) Distribution of mutations according to the MIPSS karyotype and RNA splicing genes: SF3B1, SRSF2, ZRSR2, and U2AF1. (**B**) Proportion of genetic mutations. (**C**) Proportion of variant allele frequencies (VAF). (**D**) Chord diagram showing the correlation between clinical variables and mutations. The red bar indicates a positive correlation, and the blue bar indicates a negative correlation. The red and blue bars indicate positive and negative correlations, respectively. (**E**) Decision tree classifying myelofibrosis (MF) by clinical and mutational criteria.

**Figure 2 cancers-14-04485-f002:**
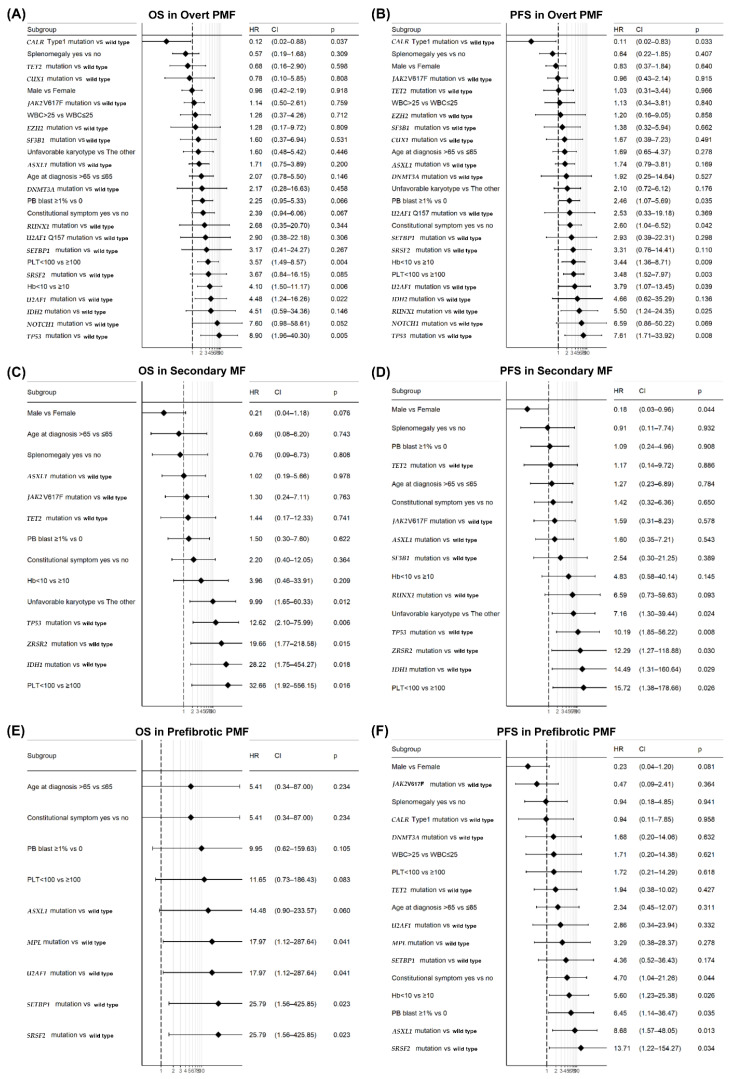
Univariate analysis of overall survival (OS) in the (**A**) primary myelofibrosis (PMF), (**C**) secondary myelofibrosis (SMF), and (**E**) prefibrotic myelofibrosis (pre-PMF) groups. Progression-free survival (PFS) in the (**B**) PMF, (**D**) SMF, and (**F**) pre-PMF groups.

**Figure 3 cancers-14-04485-f003:**
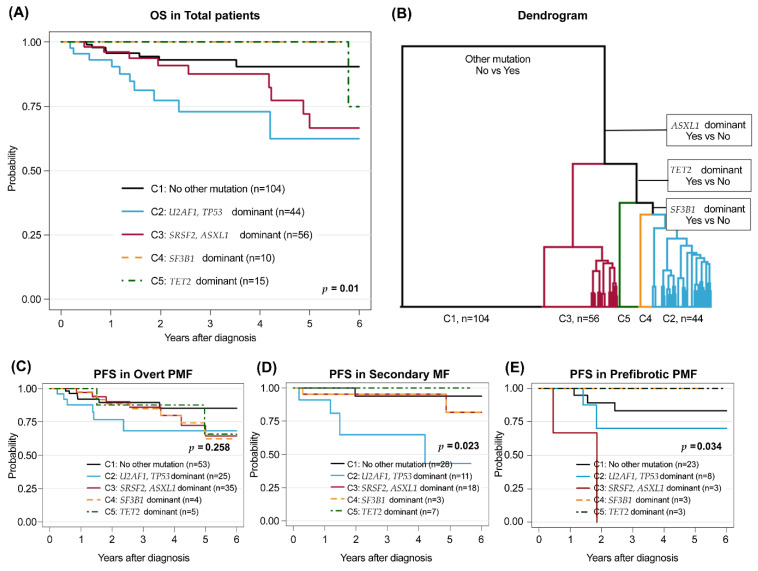
Hierarchical clustering in the overall cohort (**A**) applied to overall survival (OS) and (**B**) dendrogram. PFS by clusters in (**C**) primary myelofibrosis (PMF), (**D**) secondary myelofibrosis (SMF), and (**E**) prefibrotic myelofibrosis (pre-PMF). For detailed *p*-value between clusters is described in Appendix A.

**Figure 4 cancers-14-04485-f004:**
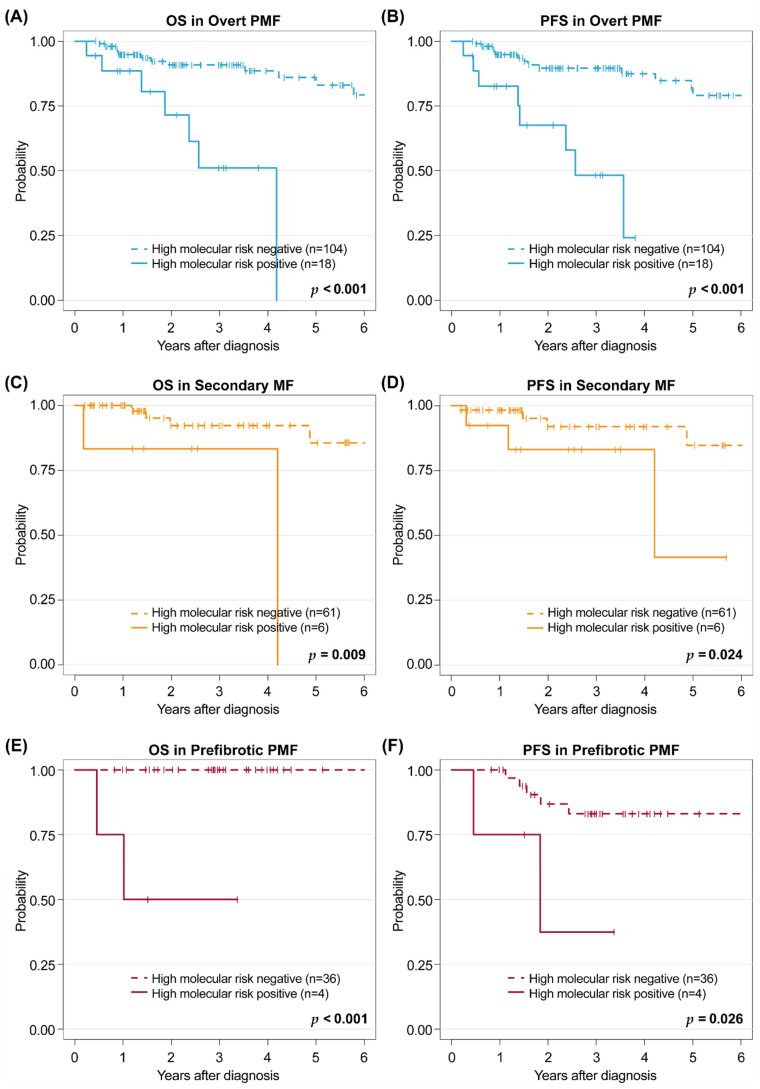
Estimation of overall survival (OS) in (**A**) primary myelofibrosis (PMF), (**C**) secondary myelofibrosis (SMF), and (**E**) prefibrotic myelofibrosis (pre-PMF), and progression-free survival (PFS) in (**B**) PMF, (**D**) SMF, and (**F**) pre-PMF after consideration of the high-risk molecular group.

**Table 1 cancers-14-04485-t001:** Baseline clinical and genetic characteristics of patients in the present study.

Variable	SMF								
	All (*n* = 229)	PPV-MF (*n* = 21)	PET-MF (*n* = 46)	SMF (*n* = 67)	PMF (*n* = 122)	Pre-PMF (*n* = 40)	SMF vs. PMF *p*	SMF vs. Pre-PMF *p*	PMF vs. Pre-PMF *p*
Age at diagnosis (years), mean ± SD	56.8 ± 12.1	62.8 ± 10.1	59.6 ± 11.9	60.6 ± 11.4	55.9 ± 11.8	53.4 ± 12.6	0.008	0.003	0.250
Sex, male, n (%)	119 (52.0)	9 (42.9)	20 (43.5)	29 (43.3)	73 (59.8)	17 (42.5)	0.042	>0.999	0.083
White blood cells (10^9^/L), mean ± SD	14.7 ± 2.6	23.0 ± 18.9	12.1 ± 9.2	15.5 ± 1.4	14.3 ± 33.6	14.4 ± 12.3	0.721	0.678	0.969
Hemoglobin (g/dL), mean ± SD	10.6 ± 2.8	11.5 ± 3.4	9.9 ± 2.0	10.4 ± 2.6	10.0 ± 2.7	12.7 ± 2.3	0.306	<0.001	<0.001
Platelet (10^9^/L), mean ± SD	442.7 ± 357.8	396.7 ± 232.4	487.5 ± 243.2	459.0 ± 241.8	350.9 ± 345.8	695.5 ± 431.6	0.013	0.002	<0.001
Peripheral blast proportion (%)	1.0 ± 1.9	1.2 ± 2.2	1.2 ± 2.3	1.2 ± 2.2	1.2 ± 1.9	0.1 ± 0.5	0.819	0.001	<0.001
Spleen size (cm) mean ± SD	15.6 ± 4.8	17.9 ± 5.5	13.7 ± 3.5	14.9 ± 4.6	16.7 ± 4.9	13.1 ± 3.2	0.018	0.017	<0.001
Constitutional symptoms, n (%)	112 (48.9)	11 (52.4)	24 (52.2)	35 (52.2)	70 (57.4)	7 (17.5)	0.598	0.001	<0.001
Allogeneic HSCT, n (%)	48 (21.0)	3 (14.3)	10 (21.7)	13 (19.4)	27 (22.1)	8 (20.0)	0.800	>0.999	0.895
Ruxolitinib exposure before HSCT, n (%)	33 (14.4)	2 (9.5)	10 (21.7)	12 (17.9)	21 (17.2)	0 (0.0)	>0.999	0.012	0.017
Treatment of non-transplant patients, n (%)	181 (79.0)	18 (85.7)	36 (78.3)	54 (80.6)	95 (77.9)	32 (80.0)	0.8	>0.999	0.895
Ruxolitinib	136 (59.6)	17 (81.0)	29 (63.0)	46 (68.7)	82 (67.8)	8 (20.0)	>0.999	<0.001	<0.001
Androgens	47 (20.6)	3 (14.3)	8 (17.4)	11 (16.4)	31 (25.6)	5 (12.5)	0.205	0.787	0.124
Hydroxyurea	91 (39.9)	14 (66.7)	26 (56.5)	40 (59.7)	31 (25.6)	20 (50.0)	<0.001	0.437	<0.001
Anagrelide	67 (29.4)	4 (19.0)	26 (56.5)	30 (44.8)	14 (11.6)	23 (57.5)	<0.001	0.283	<0.001
Thalidomide	1 (0.4)	0 (0.0)	0 (0.0)	0 (0.0)	1 (0.8)	0 (0.0)	>0.999	-	0.641
Investigational agent	13 (5.7)	2 (9.5)	2 (4.3)	4 (6.0)	5 (4.1)	4 (10.0)	0.835	0.699	0.379
Leukemic transformation, n (%)	8 (3.5)	0 (0.0)	1 (2.2)	1 (1.5)	6 (4.9)	1 (2.5)	0.429	>0.999	0.838
DIPSS Karyotype, n (%)									
Unfavorable *	17 (7.4)	2 (9.5)	4 (8.7)	6 (9.0)	11 (9.0)	0 (0.0)	>0.999	0.130	0.109
MIPSS Karyotype, n (%)							0.035	0.123	0.016
Favorable ^†^	184 (80.3)	18 (85.7)	38 (82.6)	56 (83.5)	90 (73.8)	38 (95.0)			
Intermediate	33 (14.4)	2 (9.5)	3 (6.5)	5 (7.5)	26 (21.3)	2 (5.0)			
Very High risk ^††^	12 (5.2)	1 (4.8)	5 (10.9)	6 (9.0)	6 (4.9)	0 (0.0)			
Mutation, n (%)									
*JAK2*V617F	117 (51.1)	21 (100.0)	22 (47.8)	43 (64.2)	54 (44.3)	20 (50.0)	0.014	0.215	0.653
*CALR*	60 (26.2)	0 (0.0)	18 (39.1)	18 (26.9)	35 (28.7)	7 (17.5)	0.922	0.383	0.233
Type1/like	39 (17.0)	0 (0.0)	9 (19.6)	9 (13.4)	25 (20.5)	5 (12.5)	0.320	0.377	0.534
Type2/like	16 (7.0)	0 (0.0)	8 (17.4)	8 (11.9)	7 (5.7)	1 (2.5)			
Others	5 (2.2)	0 (0.0)	1 (2.2)	1 (1.5)	3 (2.5)	1 (2.5)			
*MPL*	10 (4.4)	0 (0.0)	3 (6.5)	3 (4.5)	5 (4.1)	2 (5.0)	>0.999	>0.999	>0.999
*ASXL1*	66 (28.8)	3 (14.3)	16 (34.8)	19 (28.4)	44 (36.1)	3 (7.5)	0.361	0.020	0.001
*CBL*	5 (2.2)	1 (4.8)	0 (0.0)	1 (1.5)	4 (3.3)	0 (0.0)	0.796	>0.999	0.567
*CUX1*	4 (1.7)	0 (0.0)	0 (0.0)	0 (0.0)	4 (3.3)	0 (0.0)	0.332		0.567
*DNMT3A*	11 (4.8)	3 (14.3)	0 (0.0)	3 (4.5)	4 (3.3)	4 (10.0)	0.988	0.475	0.200
*EZH2*	5 (2.2)	1 (4.8)	2 (4.3)	3 (4.5)	2 (1.6)	0 (0.0)	0.491	0.452	>0.999
*IDH1*	4 (1.7)	1 (4.8)	1 (2.2)	2 (3.0)	2 (1.6)	0 (0.0)	0.931	0.715	>0.999
*IDH2*	4 (1.7)	1 (4.8)	0 (0.0)	1 (1.5)	1 (0.8)	2 (5.0)	>0.999	0.647	0.305
*NOTCH1*	3 (1.3)	0 (0.0)	1 (2.2)	1 (1.5)	1 (0.8)	1 (2.5)	>0.999	>0.999	0.992
*NRAS*	5 (2.2)	0 (0.0)	0 (0.0)	0 (0.0)	3 (2.5)	2 (5.0)	0.493	0.267	0.780
*RUNX1*	8 (3.5)	0 (0.0)	3 (6.5)	3 (4.5)	4 (3.3)	1 (2.5)	0.988	>0.999	>0.999
*SETBP1*	5 (2.2)	0 (0.0)	0 (0.0)	0 (0.0)	3 (2.5)	2 (5.0)	0.493	0.267	0.780
*SF3B1*	17 (7.4)	0 (0.0)	5 (10.9)	5 (7.5)	8 (6.6)	4 (10.0)	>0.999	0.922	0.709
*SRSF2*	5 (2.2)	0 (0.0)	0 (0.0)	0 (0.0)	3 (2.5)	2 (5.0)	0.493	0.267	0.780
*TET2*	28 (12.2)	4 (19.0)	5 (10.9)	9 (13.4)	12 (9.8)	7 (17.5)	0.610	0.771	0.306
*TP53*	8 (3.5)	2 (9.5)	1 (2.2)	3 (4.5)	5 (4.1)	0 (0.0)	>0.999	0.452	0.439
*U2AF1*	11 (4.8)	0 (0.0)	1 (2.2)	1 (1.5)	8 (6.6)	2 (5.0)	0.227	0.647	>0.999
*U2AF1*Q157	3 (1.3)	0 (0.0)	0 (0.0)	0 (0.0)	3 (2.5)	0 (0.0)	0.493		0.745
*ZRSR2*	4 (1.7)	1 (4.8)	1 (2.2)	2 (3.0)	1 (0.8)	1 (2.5)	0.595	>0.999	0.992

SMF, secondary myelofibrosis; PPV-MF, post-polycythemia vera myelofibrosis; PET-MF, post-essential thrombocythemia myelofibrosis; PMF, primary myelofibrosis; pre-PMF, prefibrotic, myelofibrosis; n, number; P, *p*-value; SD, standard deviation; HSCT, hematopoietic stem cell transplantation; DIPSS, Dynamic International Prognostic Scoring System; MIPSS70, Mutation-Enhanced International Prognostic Scoring System. * Unfavorable karyotype: Complex karyotype or single or tow abnormalities including +8, −7/7q-, i(17q), −5/5q-, 12p-, inv(3), or 11q23 rearrangements. ^†^ Favorable karyotype normal karyotype, sole abnormalities of 20q-, 13q-, +9, chromosome 1 translocation/duplication—sex chromosome abnormality including—Y; ^††^ Very-High-Risk karyotype: single/multiple abnormalities of −7, i(17q), inv(3)/3q21, 12p-/12p11.2, 11q-/11q23, or other autosomal trisomies not including +8/ +9 (e.g., +21, +19).

**Table 2 cancers-14-04485-t002:** Scores of risk stratification in patients with myelofibrosis.

Variable	All (*n* = 229)	SMF (*n* = 67)	PMF (*n* = 122)	pre-PMF (*n* = 40)	SMF vs. PMF *p*	SMF vs. pre-PMF *p*	PMF vs. pre-PMF *p*
IPSS Score, mean ± SD	1.8 ± 1.2	2.1 ± 1.2	2.0 ± 1.2	0.7 ± 1.0	0.574	<0.001	<0.001
IPSS Risk Group, n (%)					0.721	<0.001	<0.001
Low	44 (19.2)	6 (9.0)	15 (12.3)	23 (57.5)			
Intermediate-1	54 (23.6)	15 (22.4)	30 (24.6)	9 (22.5)			
Intermediate-2	62 (27.1)	23 (34.3)	33 (27.0)	6 (15.0)			
High	69 (30.1)	23 (34.3)	44 (36.1)	2 (5.0)			
DIPSS Score, mean ± SD	2.3 ± 1.6	2.6 ± 1.5	2.5 ± 1.5	0.8 ± 1.2	0.734	<0.001	<0.001
DPSS Risk Group, n (%)					0.813	<0.001	<0.001
Low	44 (19.2)	6 (9.0)	15 (12.3)	23 (57.5)			
Intermediate-1	77 (33.6)	24 (35.8)	42 (34.4)	11 (27.5)			
Intermediate-2	92 (40.2)	30 (44.8)	56 (45.9)	6 (15.0)			
High	16 (7.0)	7 (10.4)	9 (7.4)	0 (0.0)			
DIPSS-plus Score, mean ± SD	1.8 ± 1.4	1.9 ± 1.1	2.1 ± 1.4	0.7 ± 1.1	0.153	<0.001	<0.001
DIPSS-plus Risk Group, n (%)					0.051	<0.001	<0.001
Low	43 (18.8)	5 (7.5)	15 (12.3)	23 (57.5)			
Intermediate-1	66 (28.8)	23 (34.3)	33 (27.0)	10 (25.0)			
Intermediate-2	92 (40.2)	35 (52.2)	51 (41.8)	6 (15.0)			
High	28 (12.2)	4 (6.0)	23(18.9)	1 (2.5)			
MYSEC-PM Score, mean ± SD	12.3 ± 3.0	13.1 ± 2.5	12.5 ± 3.0	10.4 ± 2.6	0.159	<0.001	<0.001
MYSEC-PM Risk Group, n (%)					0.336	<0.001	0.002
Low	68 (29.7)	12 (17.9)	35 (28.7)	21 (52.5)			
Intermediate-1	84 (36.7)	26 (38.8)	42 (34.4)	16 (40.0)			
Intermediate-2	53 (23.1)	18 (26.9)	32 (26.2)	3 (7.5)			
High	24 (10.5)	11 (16.4)	13 (10.7)	0 (0.0)			
MIPSS70 Score, mean ± SD	3.8 ± 2.0	4.0 ± 1.6	4.4 ± 1.9	1.8 ± 1.7	0.164	<0.001	<0.001
MIPSS70 Risk Group, n (%)					0.025	<0.001	<0.001
Low	34 (14.8)	1 (1.5)	5 (4.1)	28 (70.0)			
Intermediate	111 (48.5)	46 (68.7)	59 (48.4)	6 (15.0)			
High	84 (36.7)	20 (29.9)	58 (47.5)	6 (15.0)			
MIPSS70 + Ver2 Score, mean ± SD	5.0 ± 2.9	5.1 ± 2.6	5.6 ± 3.1	2.9 ± 2.0	0.260	<0.001	<0.001
MIPSS70 + Ver2 Risk Group, n (%)					0.418	<0.001	<0.001
Very low	7 (3.1)	0 (0.0)	4 (3.3)	3 (7.5)			
Low	54 (23.6)	13 (19.4)	17 (13.9)	24 (60.0)			
Intermediate	48 (21.0)	15 (22.4)	28 (23.0)	5 (12.5)			
High	87 (38.0)	30 (44.8)	50 (41.0)	7 (17.5)			
Very high	33 (14.4)	9 (13.4)	23 (18.9)	1 (2.5)			
GIPSS Score, mean ± SD	1.4 ± 0.9	1.4 ± 0.8	1.5 ± 0.9	1.0 ± 0.6	0.410	0.208	<0.001
GIPSS Risk Group, n (%)					0.697	0.121	0.018
Low	21 (9.2)	5 (7.5)	11 (9.0)	5 (12.5)			
Intermediate-1	127 (55.5)	38 (56.7)	60 (49.2)	29 (72.5)			
Intermediate-2	54 (23.6)	17 (25.4)	32 (26.2)	5 (12.5)			
High	27 (11.8)	7 (10.4)	19 (15.6)	1 (2.5)			

SMF, secondary myelofibrosis; PMF, primary myelofibrosis; pre-PMF, prefibrotic myelofibrosis; P, *p*-value; SD, standard deviation; IPSS, International Prognostic Scoring System; MYSEC-PM, Myelofibrosis Secondary to PV and ET-Prognostic Model; GIPSS, genetically inspired prognostic scoring system.

**Table 3 cancers-14-04485-t003:** Application of risk stratification in myelofibrosis subgroups for predicting overall and progression-free survival.

Variable	Overall Survival		Progression-Free Survival *			
	HR	95% CI	*p*	HR	95% CI	*p*
pre-PMF						
Risk stratification						
IPSS, INT2 or high	-	-	>0.999	9.69	2.03, 46.1	0.004
DIPSS, INT2 or high	-	-	>0.999	11.2	2.40, 52.2	0.002
DIPSS plus, INT2 or high	-	-	>0.999	5.44	1.20, 24.7	0.028
MIPSS70, high	-	-	>0.999	6.11	1.32, 28.3	0.021
MIPSS70 +Ver2, high or very high	-	-	>0.999	3.93	0.86, 18.0	0.078
GIPSS, INT2 or high	5.93	0.37, 94.8	0.2	3.37	0.63, 18.0	0.2
MYSECPM, INT2 or high	-	-	>0.999	16.7	2.71, 103	0.002
PMF						
Risk stratification						
IPSS, INT2 or high	6.73	1.94, 23.3	0.003	6.92	2.02, 23.7	0.002
DIPSS, INT2 or high	10.9	3.06, 38.7	<0.001	7.71	2.51, 23.7	<0.001
DIPSS plus, INT2 or high	6.21	1.82, 21.3	0.004	4.64	1.57, 13.7	0.006
MIPSS70, high	6.62	2.39, 18.4	<0.001	5.41	2.11, 13.9	<0.001
MIPSS70 +Ver2, high or very high	9.05	2.10, 39.0	0.003	9.72	2.27, 41.5	0.002
GIPSS, INT2 or high	2.68	1.12, 6.43	0.027	3.1	1.32, 7.28	0.009
MYSEC-PM, INT2 or high	2.37	0.91, 6.18	0.079	2.62	1.07, 6.37	0.034
SMF						
Risk stratification						
IPSS, INT2 or high	3.63	0.42, 31.4	0.2	4.15	0.49, 34.9	0.2
DIPSS, INT2 or high	4.59	0.54, 39.4	0.2	5.48	0.66, 45.6	0.12
DIPSS plus, INT2 or high	4.46	0.52, 38.3	0.2	5.25	0.63, 43.8	0.13
MIPSS70, high	5.04	0.92, 27.6	0.062	3.33	0.74, 14.9	0.12
MIPSS70 +Ver2, high or very high	-	-	>0.999	-		>0.999
GIPSS, INT2 or high	3.26	0.59, 17.9	0.2	4.18	0.81, 21.6	0.088
MYSECPM, INT2 or high	4.18	0.75, 23.5	0.1	4.98	0.94, 26.3	0.059

* Progression refers to leukemic transformation in PMF and SMF, and leukemic transformation and fibrosis progression in pre-PMF. SMF, secondary myelofibrosis; PMF, primary myelofibrosis; pre-PMF, prefibrotic, myelofibrosis; P, *p*-value; SD, standard deviation; HR, hazard ratio; CI, confidence interval; INT2, intermediate-2.

**Table 4 cancers-14-04485-t004:** Estimation of progression-free survival using univariate and multivariate analyses of clinical and genetic factors in patients with pre-PMF.

Variable	Univariate			Multivariate Model I			Multivariate Model II		
	HR	95% CI	*p*	HR	95% CI	*p*	HR	95% CI	*p*
pre-PMF									
Clinical variable									
Age at diagnosis (years)	2.34	0.45, 12.1	0.3						
Sex, male vs. female	0.23	0.04, 1.20	0.081						
White blood cells (10^9^/L) > 25	1.71	0.20, 14.4	0.6						
Hemoglobin (g/dL) < 10	5.6	1.23, 25.4	0.026	6.30	1.02, 39.1	0.048			
Platelet (10^9^/L) < 100	1.72	0.21, 14.3	0.6						
Peripheral blast (%) > 1	6.45	1.14, 36.5	0.035						
Splenomegaly (cm)	0.94	0.18, 4.85	>0.999						
Constitutional symptom, yes	4.7	1.04, 21.3	0.044				4.47	0.75, 26.6	0.1
Genetic variable									
*JAK2*V617F	0.47	0.09, 2.41	0.4						
*CALR* Type1/like	0.94	0.11, 7.85	>0.999						
*MPL*	3.29	0.38, 28.4	0.3						
*ASXL1*	8.68	1.57, 48.1	0.013	11.4	1.59, 81.5	0.015	3.69	0.55, 24.8	0.2
*DNMT3A*	1.68	0.20, 14.1	0.6						
*RUNX1*	-	-	>0.999						
*SETBP1*	4.36	0.52, 36.4	0.2						
*SF3B1*	-	-	>0.999						
*SRSF2*	13.7	1.22, 154	0.034	2.23	0.10, 50.9	0.6	19.5	1.29, 294	0.032
*TET2*	1.94	0.38, 10.0	0.4						
*U2AF1*	2.86	0.34, 23.9	0.3						
*ZRSR2*	-	-	>0.999						

pre-PMF, prefibrotic, myelofibrosis; P, *p*-value; HR, hazard ratio; CI, confidence interval.

## Data Availability

The data presented in this study are available on request from the corresponding author. The data are not publicly available owing to privacy and ethical reasons.

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
