# Peer review of "Clinical Features, Gene Alterations, and Outcomes in Prefibrotic and Overt Primary and Secondary Myelofibrotic Patients"

_cancers, 2022, doi:10.3390/cancers14184485_

Round 1

Reviewer 1 Report

·        

Kime t al report clinal and biological findings in a large cohort of pre-MF, PMF and SMF patients. While this study is certainly of interest, conclusion are sometimes over-interpreted and following comments hould be addressed:

Introduction:

Recent reports on the topic of mutation order, pyhlogeny and time of mutations acquistion should be added (DOI: 10.1056/NEJMoa1412098; doi.org/10.1038/s41467-019-13892-x; doi.org/10.1038/s41586-021-04312-6)

 ·         Patient cohort:

The authors should add information on treatment. So far, only allogeneic HSCT is listed in Table 1. Treatment information is an important factor when evaluating PFS and OS.

 ·         Chapter results 3.3. “Correlation of genetic and clinical features”:

The authors performed extensive association tests, none of them corrected for multiple testing. I strongly recommend to reduce tests to a meaningful number. Association tests should be restricted to genes with a mutation prevalence of at least 5%. There is no interest at all to report “significant findings” such as “In the PMF group, we found that the IDH2 mutations correlated with higher leukocyte counts, whereas the U2AF1 and RUNX1 mutations associated with thrombocytopenia”. These data are based on 1-3% of patients in the respective groups and statistical robustness is more than questionable.

 ·         Chapter results 3.6. “Genomic subgroups in myelofibrosis by nondriver mutations”:

The authors perform interesting cluster analysis but mentioned p-values are unclear to me. As 5 groups are compared per test, not an overall p-value but intergroup comparison p-values showed be given (e.g. p-value for C2 vs C1 or C2 vs all others)

 ·         Chapter results 3.7. “Proposal of high-risk mutation groups predicting survival outcomes”:

Please clarify which mutations you classified as ARCH/CHIP mutations. Currently, I understand that you grouped SRSF2, U2AF1, SF3B1, IDH1/2, and EZH2 mutations as “ARCH/CHIP”. However, this wrong and the mentioned article by Bartels et al () reads as follows: “Frequent ARCH/CHIP-associated mutations (TET2, ASXL1, and DNMT3A) demonstrable at presentation were not connected with fibrotic progression. However, mutations which are rarely found in ARCH/CHIP (SRSF2, U2AF1, SF3B1, IDH1/2, and EZH2) were present in 24.7% of cases with later development of fibrosis”. Please clarify and adopt!

 ·         Discussion:

The discussion is too long and repetitive, and again, genes such as U2AF1 are intensively discussed but findings are based on 3% of the entire cohort!

Minor comments:

·         The authors must use the word “wild-type” instead of “wild” in every figure, figure legend, table and text.

·         Gene names should be written in italics in all figures and tables

Author Response

All replies and tables or figures for response are included in the file.

  • Comments from Reviewer 1

    We profoundly reviewed and discussed our manuscript. We looked over the comments and changes could be achieved. Despite some weaknesses in our study, we believe that our findings will be dedicated to inspire point of views on MF. 

    Comment 1: Introduction: Recent reports on the topic of mutation order, pyhlogeny and time of mutations acquistion should be added (DOI: 10.1056/NEJMoa1412098; doi.org/10.1038/s41467-019-13892-x; doi.org/10.1038/s41586-021-04312-6)

    -  We appreciate your comment. As a result, we decided to add sentences and mentioned the references.

    After revision: 1. Introduction part, below is modified (in blue) (Line 76)

    As the studies of mutation order and acquisition accumulates, more interest is in correlation with the clinical course of myelofibrosis and genomic status. Due to clonal evolution during disease progression and various latencies among the genes, order of mutations was different among the patients. Furthermore, mutations order affects the presentation of clinical course and drug susceptibility [14-16].

    Comment 2: Patient cohort:The authors should add information on treatment. So far, only allogeneic HSCT is listed in Table 1. Treatment information is an important factor when evaluating PFS and OS.

    - As per your suggestion, all authors agreed on the need for a treatment information on Table1. We had described on revised Table.

    After revision: 3.1. Clinical and cytogenetic features of patients with prefibrotic, overt PMF, and secondary MF part, below is modified (in blue) (Line 153)

    Variable

    SMF

    All

    (n = 229)

    PPV-MF

    (n = 21)

    PET-MF

    (n = 46)

    SMF

    (n=67)

    PMF

    (n = 122)

    pre-PMF

    (n = 40)

    SMF

    vs.

    PMF

    P

    SMF

    vs.

    pre-PMF

    P

    PMF

    vs.

    pre-PMF

    P

    Age at diagnosis (years), mean ± SD

    56.8 ± 12.1

    62.8 ± 10.1

    59.6 ± 11.9

    60.6 ± 11.4

    55.9 ± 11.8

    53.4 ± 12.6

    .008

    .003

    .250

    Sex, male, n (%)

    119 (52.0)

    9 (42.9)

    20 (43.5)

    29 (43.3)

    73 (59.8)

    17 (42.5)

    .042

    >.999

    .083

    White blood cells (109/L), mean ± SD

    14.7 ± 2.6

    23.0 ± 18.9

    12.1 ± 9.2

    15.5 ± 1.4

    14.3 ± 33.6

    14.4 ± 12.3

    .721

    .678

    .969

    Hemoglobin (g/dL), mean ± SD

    10.6 ± 2.8

    11.5 ± 3.4

    9.9 ± 2.0

    10.4 ± 2.6

    10.0 ± 2.7

    12.7 ± 2.3

    .306

    <.001

    <.001

    Platelet (109/L), mean ± SD

    442.7 ± 357.8

    396.7 ± 232.4

    487.5 ± 243.2

    459.0 ± 241.8

    350.9 ± 345.8

    695.5 ± 431.6

    .013

    .002

    <.001

    Peripheral blast

    proportion (%)

    1.0 ± 1.9

    1.2 ± 2.2

    1.2 ± 2.3

    1.2 ± 2.2

    1.2 ± 1.9

    0.1 ± 0.5

    .819

    .001

    <.001

    Spleen size (cm) mean ± SD

    15.6 ± 4.8

    17.9 ± 5.5

    13.7 ± 3.5

    14.9 ± 4.6

    16.7 ± 4.9

    13.1 ± 3.2

    .018

    .017

    <.001

    Constitutional

    symptoms, n (%)

    112 (48.9)

    11 (52.4)

    24 (52.2)

    35 (52.2)

    70 (57.4)

    7 (17.5)

    .598

    .001

    <.001

    Allogenic HSCT, n (%)

    48 (21.0)

    3 (14.3)

    10 (21.7)

    13 (19.4)

    27 (22.1)

    8 (20.0)

    .800

    >.999

    .895

    Ruxolitinib exposure before HSCT, n (%)

    33 (14.4)

    2 ( 9.5)

    10 (21.7)

    12 (17.9)

    21 (17.2)

    0 ( 0.0%)

    >.999

    .012

    .017

    Treatment of non-transplant patients, n (%)

    181 (79.0)

    18 (85.7)

    36 (78.3)

    54 (80.6)

    95 (77.9)

    32 (80.0)

    0.8

    >.999

    .895

    Ruxolitinib

    136 (59.6)

    17 (81.0)

    29 (63.0)

    46 (68.7)

    82 (67.8)

    8 (20.0)

    >.999

    <.001

    <.001

    Androgens

    47 (20.6)

    3 (14.3)

    8 (17.4)

    11 (16.4)

    31 (25.6)

    5 (12.5)

    0.205

    .787

    .124

    Hydroxyurea

    91 (39.9)

    14 (66.7)

    26 (56.5)

    40 (59.7)

    31 (25.6)

    20 (50.0)

    <.001

    .437

    <.001

    Anagrelide

    67 (29.4)

    4 (19.0)

    26 (56.5)

    30 (44.8)

    14 (11.6)

    23 (57.5)

    <.001

    .283

    <.001

    Thalidomide

    1 (0.4)

    0 ( 0.0)

    0 ( 0.0)

    0 ( 0.0)

    1 ( 0.8)

    0 ( 0.0)

    >.999

    -

    .641

    Investigational agent

    13 (5.7)

    2 ( 9.5)

    2 ( 4.3)

    4 ( 6.0)

    5 ( 4.1)

    4 (10.0)

    .835

    .699

    .379

    Leukemic transformation, n (%)

    8 (3.5)

    0 (0.0)

    1 (2.2)

    1 (1.5)

    6 (4.9)

    1 (2.5)

    .429

    >.999

    .838

    DIPSS Karyotype, n (%)

    *Unfavorable

    17 (7.4)

    2 (9.5)

    4 (8.7)

    6 (9.0)

    11 (9.0)

    0 (0.0)

    >.999

    .130

    .109

    MIPSS Karyotype, n (%)

    .035

    .123

    .016

    † Favorable

    184 (80.3)

    18 (85.7)

    38 (82.6)

    56 (83.5)

    90 (73.8)

    38 (95.0)

       Intermediate

    33 (14.4)

    2 (9.5)

    3 (6.5)

    5 (7.5)

    26 (21.3)

    2 (5.0)

    †† Very High risk

    12 (5.2)

    1 (4.8)

    5 (10.9)

    6 (9.0)

    6 (4.9)

    0 (0.0)

    Mutation, n (%)

    JAK2V617F

    117 (51.1)

    21 (100.0)

    22 (47.8)

    43 (64.2)

    54 (44.3)

    20 (50.0)

    .014

    .215

    .653

    CALR

    60 (26.2)

    0 (0.0)

    18 (39.1)

    18 (26.9)

    35 (28.7)

    7 (17.5)

    .922

    .383

    .233

        Type1/like

    39 (17.0)

    0 (0.0)

    9 (19.6)

    9 (13.4)

    25 (20.5)

    5 (12.5)

    .320

    .377

    .534

        Type2/like

    16 (7.0)

    0 (0.0)

    8 (17.4)

    8 (11.9)

    7 (5.7)

    1 (2.5)

        Others

    5 (2.2)

    0 (0.0)

    1 (2.2)

    1 (1.5)

    3 (2.5)

    1 (2.5)

    MPL

    10 (4.4)

    0 (0.0)

    3 (6.5)

    3 (4.5)

    5 (4.1)

    2 (5.0)

    >.999

    >.999

    >.999

    ASXL1      

    66 (28.8)

    3 (14.3)

    16 (34.8)

    19 (28.4)

    44 (36.1)

    3 (7.5)

    .361

    .020

    .001

    CBL        

    5 (2.2)

    1 (4.8)

    0 (0.0)

    1 (1.5)

    4 (3.3)

    0 (0.0)

    .796

    >.999

    .567

    CUX1       

    4 (1.7)

    0 (0.0)

    0 (0.0)

    0 (0.0)

    4 (3.3)

    0 (0.0)

    .332

    .567

    DNMT3A     

    11 (4.8)

    3 (14.3)

    0 (0.0)

    3 (4.5)

    4 (3.3)

    4 (10.0)

    .988

    .475

    .200

    EZH2       

    5 (2.2)

    1 (4.8)

    2 (4.3)

    3 (4.5)

    2 (1.6)

    0 (0.0)

    .491

    .452

    >.999

    IDH1       

    4 (1.7)

    1 (4.8)

    1 (2.2)

    2 (3.0)

    2 (1.6)

    0 (0.0)

    .931

    .715

    >.999

    IDH2       

    4 (1.7)

    1 (4.8)

    0 (0.0)

    1 (1.5)

    1 (0.8)

    2 (5.0)

    >.999

    .647

    .305

    NOTCH1     

    3 (1.3)

    0 (0.0)

    1 (2.2)

    1 (1.5)

    1 (0.8)

    1 (2.5)

    >.999

    >.999

    .992

    NRAS       

    5 (2.2)

    0 (0.0)

    0 (0.0)

    0 (0.0)

    3 (2.5)

    2 (5.0)

    .493

    .267

    .780

    RUNX1      

    8 (3.5)

    0 (0.0)

    3 (6.5)

    3 (4.5)

    4 (3.3)

    1 (2.5)

    .988

    >.999

    >.999

    SETBP1     

    5 (2.2)

    0 (0.0)

    0 (0.0)

    0 (0.0)

    3 (2.5)

    2 (5.0)

    .493

    .267

    .780

    SF3B1      

    17 (7.4)

    0 (0.0)

    5 (10.9)

    5 (7.5)

    8 (6.6)

    4 (10.0)

    >.999

    .922

    .709

    SRSF2      

    5 (2.2)

    0 (0.0)

    0 (0.0)

    0 (0.0)

    3 (2.5)

    2 (5.0)

    .493

    .267

    .780

    TET2       

    28 (12.2)

    4 (19.0)

    5 (10.9)

    9 (13.4)

    12 (9.8)

    7 (17.5)

    .610

    .771

    .306

    TP53       

    8 (3.5)

    2 (9.5)

    1 (2.2)

    3 (4.5)

    5 (4.1)

    0 (0.0)

    >.999

    .452

    .439

    U2AF1      

    11 (4.8)

    0 (0.0)

    1 (2.2)

    1 (1.5)

    8 (6.6)

    2 (5.0)

    .227

    .647

    >.999

    U2AF1Q157

    3 (1.3)

    0 (0.0)

    0 (0.0)

    0 (0.0)

    3 (2.5)

    0 (0.0)

    .493

    .745

    ZRSR2      

    4 (1.7)

    1 (4.8)

    1 (2.2)

    2 (3.0)

    1 (0.8)

    1 (2.5)

    .595

    >.999

    .992

    Comment 3: Chapter results 3.3. “Correlation of genetic and clinical features”:The authors performed extensive association tests, none of them corrected for multiple testing. I strongly recommend to reduce tests to a meaningful number. Association tests should be restricted to genes with a mutation prevalence of at least 5%. There is no interest at all to report “significant findings” such as “In the PMF group, we found that the IDH2 mutations correlated with higher leukocyte counts, whereas the U2AF1 and RUNX1 mutations associated with thrombocytopenia”. These data are based on 1-3% of patients in the respective groups and statistical robustness is more than questionable.

    - We sincerely agree with your concern. Despite the statistical relevance, the results could confuse the readers. So, we deleted descriptions of the genes less than 5% among the subgroups.

    After revision: Chapter results 3.3. Correlation of genetic and clinical features part, below is modified (in blue) (Line 199)

    We further analyzed the correlation of these gene mutations with clinical characteris-tics. In the PMF group, we found that the IDH2 mutations correlated with higher leukocyte counts, whereas the U2AF1 and RUNX1 mutations associated with thrombocytopenia. In the SMF group, mutations in JAK2 and CALR, DNMT3A, and IDH2 correlated with higher leukocyte counts. In addition, mutations in DNMT3A, IDH1, and IDH2 correlated with anemia. Furthermore, we found that genetic aberrant TP53 positively correlated with thrombocytopenia, whereas the U2AF1 mutations negatively correlated with thrombocytopenia. In the pre-PMF group, we noticed that higher leukocyte counts were positively correlated with NOTCH1, whereas negatively correlated with IDH2 and ASXL1 mutations. The peripheral blast was positively correlated with ZRSR2, but negatively correlated with the ASXL1 mutations. Anemia was positively correlated with the JAK2 and DNMT3A and ZRSR2 mutations.  whereas thrombocytopenia was positively correlated with U2AF1 and MPL was negatively correlated with the TET2 mutations. Thrombocytopenia was positively correlated with U2AF1 and MPL whereas thrombocytopenia was negatively correlated with the TET2 mutations (Figure 1D).

    Comment 4: Chapter results 3.6. “Genomic subgroups in myelofibrosis by nondriver mutations”: The authors perform interesting cluster analysis but mentioned p-values are unclear to me. As 5 groups are compared per test, not an overall p-value but intergroup comparison p-values showed be given (e.g. p-value for C2 vs C1 or C2 vs all others)

    - We agree with the concerns from your comments, so we decided to add the supplementary table and comments on manuscript.

    After revision: 3.6. Genomic subgroups in myelofibrosis by nondriver mutations part, below is modified (in blue) (Line 296)

    Figure 3. Hierarchical clustering in the overall cohort (A) applied to overall survival (OS) and (B) dendrogram. PFS by clusters in (C) primary myelofibrosis (PMF), (D) secondary myelofibrosis (SMF), and (E) prefibrotic myelofibrosis (pre-PMF).  For the detailed p-values between clusters are described in supplementary Table S3.

    Whole cohort

    C1 vs The other clusters

    C2 vs The other clusters

    C3 vs The other clusters

    C1 vs C2

    C1 vs C3

    C2 vs. C3

    Whole cohort

    OS, p-value

    .035

    .002

    .386

    .002

    .079

    .144

    PFS, p-value

    .033

    .002

    .421

    .002

    .088

    .140

    PMF

    OS, p-value

    .188

    .163

    .468

    .162

    .226

    .492

    PFS, p-value

    .097

    .084

    .677

    .073

    .224

    .335

    SMF

    OS, p-value

    .239

    .001

    .411

    .008

    .946

    .016

    PFS, p-value

    .157

    .001

    .872

    .008

    .449

    .066

    pre-PMF

    OS, p-value

    .107

    .317

    .013

    .114

    .006

    .522

    PFS, p-value

    .458

    .522

    .002

    .468

    .005

    .152

    Comment 5: Chapter results 3.7. “Proposal of high-risk mutation groups predicting survival outcomes”:Please clarify which mutations you classified as ARCH/CHIP mutations. Currently, I understand that you grouped SRSF2, U2AF1, SF3B1, IDH1/2, and EZH2 mutations as “ARCH/CHIP”. However, this wrong and the mentioned article by Bartels et al () reads as follows: “Frequent ARCH/CHIP-associated mutations (TET2, ASXL1, and DNMT3A) demonstrable at presentation were not connected with fibrotic progression. However, mutations which are rarely found in ARCH/CHIP (SRSF2, U2AF1, SF3B1, IDH1/2, and EZH2) were present in 24.7% of cases with later development of fibrosis”. Please clarify and adopt!

    -Thank you for clarifying the sentences in ARCH/CHIP. Now we considered what made it confusing and then changed the sentences.

    After revision: 3.7. Proposal of high-risk mutation groups predicting survival outcomes part, below is modified (in blue) (Line 300)

    Among the ARCH/CHIP-associated mutations, the group of mutations was associated with later progression. that is, mutations in SRSF2, U2AF1, SF3B1, IDH1/2, and EZH2.

    Comment 6: Discussion: The discussion is too long and repetitive, and again, genes such as U2AF1 are intensively discussed but findings are based on 3% of the entire cohort!

    - We appreciate your comments, we tried to revise the discussion section more clearly and removed repetitive sentences. we hoped your worries came somewhat resolved.

    After revision: Disscussion part, below is modified (in blue) (Line 318, Line 353)

    (Line 318) In this study, we showed the clinical and genetic characteristics of patients with pre-fibrotic, overt PMF, and SMF. Based on the recent advances in genomic technologies, we attempted to predict the poor outcome in MPNs using molecular indicators. In clinical practice, these factors can be used in deciding early intervention strategies in order to minimize disease-related mortality. Although several previous studies have compared pre-PMF versus ET [27], pre-PMF versus PMF [6,28], and PMF versus SMF [29], to date, no other study except the present study has compared the clinical and genetic characteristics of patients with pre-PMF versus overt PMF and secondary MF.  A comparison of the clinical features between pre-PMF and PMF showed that patients with pre-PMF were characterized by higher hemoglobin levels and platelet counts, whereas they were less frequently diagnosed with increased peripheral blood blasts, symptoms, and extensive splenomegaly. These findings are consistent with those of a previous study [6]. Our findings on clinical feature differences between PMF and pre-PMF are consistent with those of a previous study [24]. Clinical features such as anemia, thrombocytopenia, high blast count, symptoms, large splenomegaly, and unfavorable karyotype were fre-quently observed in patients with PMF than those with pre-PMF. Similar to the study of Guglielmelli et al., we identified that AXSL1 showed a higher proportion in the patients with PMF but SRSF2, IDH1/2, and EZH2 showed no statistical difference.

    (Line 353) In the present study, the overt PMF and pre-PMF groups showed a positive correlation with U2AF1 mutations and thrombocytopenia; however, this correlation was not observed in the SMF group. This suggests that with the progression of fibrosis, U2AF1 mutations in PMF may result in thrombocytopenia. Moreover, U2AF1 mutations were associated with low overall survival in PMF and pre-PMF

    Comment 7: The authors must use the word “wild-type” instead of “wild” in every figure, figure legend, table and text. Gene names should be written in italics in all figures and tables

    - As your comments we reflect what you point. And revised on the figure, figure legend, table and text.

Reviewer 2 Report

The authors compare different prognostic scoring systems in myelofibrosis and apply them to different categories of patients. They also weigh the different molecular mutations and give interesting insights for the future use of these markers.

Major suggestions:

Row 70-71: the statement is not true. There are data in early MF in the paper: Guglielmelli P, Pacilli A, Rotunno G, Rumi E, Rosti V, Delaini F, Maffioli M, Fanelli T, Pancrazzi A, Pietra D, Salmoiraghi S, Mannarelli C, Franci A, Paoli C, Rambaldi A, Passamonti F, Barosi G, Barbui T, Cazzola M, Vannucchi AM; AGIMM Group. Presentation and outcome of patients with 2016 WHO diagnosis of prefibrotic and overt primary myelofibrosis. Blood. 2017 Jun 15;129(24):3227-3236. doi: 10.1182/blood-2017-01-761999. Epub 2017 Mar 28. PMID: 28351937;

Row 187: remove JAK2 and CARL as they are present in 62 out of 67 SMF patients and thus the statistical significance cannot be underlined.

Minor suggestions:

Row 56: use "death" instead of "mortality";

Row 70: remove "only";

Row 397-398: add the acronym GIPSS as it is better known by the readers.

Author Response

Comments from Reviewer 2

We appreciate your interest in our study, and we effort to satisfy your comments. We hope your intention had reflected.

Comment 1: Row 70-71: the statement is not true. There are data in early MF in the paper: Guglielmelli P, Pacilli A, Rotunno G, Rumi E, Rosti V, Delaini F, Maffioli M, Fanelli T, Pancrazzi A, Pietra D, Salmoiraghi S, Mannarelli C, Franci A, Paoli C, Rambaldi A, Passamonti F, Barosi G, Barbui T, Cazzola M, Vannucchi AM; AGIMM Group. Presentation and outcome of patients with 2016 WHO diagnosis of prefibrotic and overt primary myelofibrosis. Blood. 2017 Jun 15;129(24):3227-3236. doi: 10.1182/blood-2017-01-761999. Epub 2017 Mar 28. PMID: 28351937;

- As your comment, we added the reference and corrected the sentence.

After revision: 2.2. Molecular and cytogenetic studies part, below is modified (in blue) (Line 70)

To date, only sufficiently powered findings have been obtained in patients with PMF, and only a few studies have focused on secondary MF. Although there was a study on early MF [6], comparison studies between pre-PMF, overt PMF, and SMF data were limited.

Comment 2: Row 187: remove JAK2 and CARL as they are present in 62 out of 67 SMF patients and thus the statistical significance cannot be underlined.

- As you mentioned, despite the statistical relevance, the results could confuse the readers. So, we removed JAK2 and CARL. In addition, as the request of Reviewer 1, we deleted descriptions of the genes less than 5% among the subgroups.

After revision: 3.3. Correlation of genetic and clinical features  part, below is modified (in blue) (Line 200)

In the SMF group, mutations in JAK2 and CALR, DNMT3A, and IDH2 correlated with higher leukocyte counts.

 Comment 3: Row 56: use "death" instead of "mortality"; Row 70: remove "only"; Row 397-398: add the acronym GIPSS as it is better known by the readers.

- As per your comment, we revised the errors in our manuscript.

After revision: Introduction and disscussion part, below is modified (in blue) (Line 56) (Line 402)

(Line 56) The progression of myelofibrosis might lead to various pathological conditions, in-cluding thrombosis, infection, leukemic transformation, and eventually death.

(Line 402) However, a widely known GIPSS in overt PMF could not correctly predict the outcome in SMF and pre-PMF

Reviewer 3 Report

Authors present a mainly descriptive article displaying differences between prePMF overt-PMF and SMF patients. The study cohort is limited but represents a great effort for a single center. I believe that the novelty is scarce, but the molecular characterization of patients included in the present study is appealing.

I have some concerns that needs to be clarified.

Major:

- Can the authors provide a list of genes analyzed with NGS in a supplementary table?

- In Table 1 the reported comparisons are SMFvs.PMF SMFvs.pre-PMF and PMFvs.pre-PMF, nevertheless there is no cumulative column for SMF samples because PPV-MF and PET-MF are reported separately. For this reason the reader can't directly observe the data that were compared by the autors. Would it be possible to include one column reporting cumulative statistics for the SMF group?

- in line 155 authors refer to figure 1 but it is missing.

- in line 164 authors should refer to Table 1.

- Considering karyotype classification (lines 141 - 146) it is not immediately clear what the authors refer to. When they say, in text and table, DIPSS karyotype and MIPSS karyotype they should specify that they are referring to the different strategies adopted through the years for the classification of karyotype abnormalities. would it be possible to state it more clearly in the main text or in table 1 that DIPSS karyotype and MIPSS karyotype are the classification methods adopted by the two scoring systems?

- lines 172 - 182: authors highlight that there is a negative correlation between JAK2 CALR and MPL mutations in SMP PMF and pre-PMF patients. This is not a novel result, it has been widely described in literature before. They should not underline this result, they must state that this is not a novel result in line with alrready published literature. JAK2 CALr and MPL mutations are considered as driver mutations since they are mutually exclusive in MPN patients.  In supplementary figure S1 A B C D the gene symbols can't be read therefore the plots can't be understood. Please write gene symbols on the top of columns and on the side of rows. why are there some crosses over dots and numbers in panel B?

- lines 184 - 206: there is no figure 1 therefore it is not possible to make any comment on this.

- in Table 3 it is not clear to me why for many analysis there is no value reported for HR or CI. What does "-" stands for? In my opinion, it is quite surprising that MYSEC-PM is not able to efficently stratify SMF patients in different risk categories since it was designed specifically for this class of patients. Can the authors comment on this?

- in line 217 do the authors refer to "overt" myelofibrosis?

- I can't understand what the authors mean in line 219 - 220. Can they clarify their conclusion?

- line 237 - 240: authors say that 48 patients underwent allogenic HSCT. Were these patients considered for survival analysis? If yes, why they were not excluded? In my opinion these patients should have been excluded from the present study, at least for survival analysis, or authors should have consider the date of transplantation as censoring time for these patients since this treatment modality may generate confunding results. Indeed in their pubblication Passamonti et al. (https://doi.org/10.1182/blood-2009-09-245837adopted the second strategy while Gangat et al (DOI: 10.1200/JCO.2010.32.2446) performed both analyses in order to exclude the introduction of any bias. Can the author perform analysis considering the time of transplantation as censoring time?

- paragraph 3.6: Grinfeld and colleagues in 2018 proposed a genetic classification for patients affected by Philadelphia negative myeloproliferative neoplasms (doi: 10.1056/NEJMoa1716614). Their classification included 6 cathegories based on the presence/absence of specific mutations and and chromosomal abnormalities. Can the authors verify whether classification proposed by Grinfeld and colleagues is effective also in the presented cohort.

- In last results' paragraph why did the authors decided to not consider the HMR mutations identified for PMF patients (i.e. ASXL1, EZH2, IDH1/2, SRSF2, U2AF1) that they cyted in intruduction? Can they provide any result about their relevance as high risk mutations not only in overt PMF but also in SMF and pre-PMF? Moreover in dscussion lines 355 - 357 authors say that their results are not completely in line with those reported by Vannucchi et al. Vannucchi et al. reported, in their pubblication that the presence of at least one mutation affecting HMR genes is an independent risk factor for survival in both cohorts included in their pubblication. Furthermore the same mutations were included in the most recent scoring systems confirming their prognostic relevance. Can the authors analyze the impact of the presence of HMR mutations in the studied cohort considering HMR mutations as whole and not independently? 

- In figure 4 authors show the negative impact of wild type SF2B1 or mutations in SRSF2, RUNX1, U2AF1, ASXL1, and TP53 on survival and progression in the study cohort. Did they perform a multivariate analysis? do this "high molecular risk" profile represent a risk factor for survival independent from current prognostic systems? 

Minor:

- line 63 "myeloproliferations" should be "myeloproliferation"

- line 136: why PPV appears before pre-PMF? The number of pre-PMF cases is greater then those of PPV-MF.

- line 209: remove "T"

- lines 236 -237: substitute "With" with "within". 

- line 360 correct "Grinfeld"

Author Response

All replies and tables or figures for response are included in the file.

Comments from Reviewer 3

We appreciate your feedback and try to respond to all your comments. Although some points could still confuse you, we have tried our best to answer your questions and revise the relevant sections of the manuscript.

Comment 1: Can the authors provide a list of genes analyzed with NGS in a supplementary table?

- We added List of target genes in the NGS panel on the supplementary Table S1.

After revision: 2.2. Molecular and cytogenetic studies part, below is modified (in blue) (Line 102)

The SM panel contains 87 genes frequently found mutated in patients with MPN (Supplementary Table S1).

Target genes

Interval

Number of regions

Size (base pairs)

ABCA12

chr2:215797358-216002931

54

7929

ABL1

chr9:133589707-133761070

12

3529

ASXL1

chr20:30946579-31025141

17

4720

ATM

chr11:108098352-108236235

62

9171

ATRX

chrX:76763829-77041487

36

7543

ATXN7L1

chr7:105248299-105517004

15

3132

BCOR

chrX:39909169-39937182

15

5348

BRAF

chr7:140426294-140624503

21

2379

BRCC3

chrX:154299803-154348425

11

1021

CALR

chr19:13049494-13054795

10

1294

CBL

chr11:119077128-119170491

16

2721

CBLB

chr3:105377814-105588232

20

3079

CD101

chr1:117544440-117576723

9

3066

CEBPA

chr19:33792244-33793425

1

1182

CREBBP

chr16:3777719-3929917

31

7368

CSF1R

chr5:149433632-149465990

22

3003

CSF3R

chr1:36931697-36945097

16

2698

CTCF

chr16:67644736-67671775

10

2184

CUX1

chr7:101459311-101926382

34

5433

DNMT1

chr19:10244343-10311559

43

5292

DNMT3A

chr2:25457148-25536853

25

2888

EGFR

chr7:55086971-55273310

31

4084

EP300

chr22:41489009-41574960

31

7245

ERG

chr21:39739557-39947624

12

1764

ETV6

chr12:11803062-12044535

10

1443

EZH2

chr7:148504738-148544390

21

2456

FBXW7

chr4:153244033-153332955

14

2618

FLT3

chr13:28578189-28674647

25

3004

GATA1

chrX:48649517-48652675

5

1346

GATA2

chr3:128199862-128205874

5

1443

GNAS

chr20:57415162-57485884

17

4096

HIPK2

chr7:139257673-139477422

16

3708

IDH1

chr2:209101803-209116275

8

1248

IDH2

chr15:90627498-90645622

11

1359

INVS

chr9:102866804-103062956

17

3395

IRF1

chr5:131819643-131825170

9

978

JAK2

chr9:5021988-5126791

23

3399

KDM2B

chr12:121867919-122018816

29

4276

KDM6A

chrX:44732798-44970656

31

4470

KIT

chr4:55524182-55604723

21

2931

KMT2A

chr11:118307228-118392887

38

12082

KMT2D

chr12:49415563-49449107

55

16662

KRAS

chr12:25362729-25398318

6

708

LAMB4

chr7:107664484-107763609

35

5507

MECOM

chr3:168802697-169381160

20

3816

MET

chr7:116335811-116436178

21

4359

MLL3

chr7:151833917-152132871

62

15030

MLL5

chr7:104681400-104753780

27

5743

MN1

chr22:28146903-28196531

2

3963

MPL

chr1:43803520-43818443

12

1993

NCOR2

chr12:124809948-124979797

49

7734

NF1

chr17:29422226-29705949

63

9011

NLRP1

chr17:5405134-5487277

18

4493

NOTCH1

chr9:139390523-139440238

34

7668

NPM1

chr5:170814953-170837569

12

894

NRAS

chr1:115251156-115258781

4

570

NRD1

chr1:52254908-52344287

34

3696

NUP98

chr11:3692612-3803347

35

5567

OCA2

chr15:28000534-28327020

24

2573

PDGFRA

chr4:55106220-55161439

24

3450

PHF12

chr17:27233201-27278622

17

3519

PHF6

chrX:133511648-133559360

9

1207

PRPF40B

chr12:50017374-50037975

26

2682

PRPF8

chr17:1553953-1587865

42

7151

PTPN11

chr12:112856916-112942568

16

1822

RAD21

chr8:117859739-117878968

13

1896

RAD50

chr5:131893017-131978781

27

4211

RINT1

chr7:105172763-105207758

16

2417

ROBO1

chr3:78648063-79639061

34

5223

ROBO2

chr3:75986645-77695209

32

4862

RUNX1

chr21:36164432-36421196

11

1584

RUNX1T1

chr8:92972470-93115112

20

2350

SETBP1

chr18:42281312-42643663

6

4980

SF3A1

chr22:30730583-30752781

16

2382

SF3B1

chr2:198257027-198299723

27

4045

SMC1A

chrX:53407024-53449549

26

3882

SMC3

chr10:112327575-112364060

29

3654

SRSF2

chr17:74732243-74733242

2

666

STAG2

chrX:123156478-123234447

34

3861

TET1

chr10:70332096-70451571

11

6411

TET2

chr4:106111627-106197676

10

6165

TP53

chr17:7565257-7579912

14

1378

TP53BP1

chr15:43699581-43785241

31

6130

U2AF1

chr21:44513212-44527604

9

790

U2AF2

chr19:56166471-56185434

14

1541

WT1

chr11:32410604-32456891

11

1568

ZRSR2

chrX:15808619-15841365

12

1690

Comment 2: In Table 1 the reported comparisons are SMFvs.PMF SMFvs.pre-PMF and PMFvs.pre-PMF, nevertheless there is no cumulative column for SMF samples because PPV-MF and PET-MF are reported separately. For this reason, the reader can't directly observe the data that were compared by the autors. Would it be possible to include one column reporting cumulative statistics for the SMF group?

- We understand your point of view. So, we added the SMF group in Table1.

After revision: Table1, below is modified (in blue) (Line 153)

Variable

SMF

All

(n = 229)

PPV-MF

(n = 21)

PET-MF

(n = 46)

SMF

(n=67)

PMF

(n = 122)

pre-PMF

(n = 40)

SMF

vs.

PMF

P

SMF

vs.

pre-PMF

P

PMF

vs.

pre-PMF

P

Age at diagnosis (years), mean ± SD

56.8 ± 12.1

62.8 ± 10.1

59.6 ± 11.9

60.6 ± 11.4

55.9 ± 11.8

53.4 ± 12.6

.008

.003

.250

Sex, male, n (%)

119 (52.0)

9 (42.9)

20 (43.5)

29 (43.3)

73 (59.8)

17 (42.5)

.042

>.999

.083

White blood cells (109/L), mean ± SD

14.7 ± 2.6

23.0 ± 18.9

12.1 ± 9.2

15.5 ± 1.4

14.3 ± 33.6

14.4 ± 12.3

.721

.678

.969

Hemoglobin (g/dL), mean ± SD

10.6 ± 2.8

11.5 ± 3.4

9.9 ± 2.0

10.4 ± 2.6

10.0 ± 2.7

12.7 ± 2.3

.306

<.001

<.001

Platelet (109/L), mean ± SD

442.7 ± 357.8

396.7 ± 232.4

487.5 ± 243.2

459.0 ± 241.8

350.9 ± 345.8

695.5 ± 431.6

.013

.002

<.001

Peripheral blast

proportion (%)

1.0 ± 1.9

1.2 ± 2.2

1.2 ± 2.3

1.2 ± 2.2

1.2 ± 1.9

0.1 ± 0.5

.819

.001

<.001

Spleen size (cm) mean ± SD

15.6 ± 4.8

17.9 ± 5.5

13.7 ± 3.5

14.9 ± 4.6

16.7 ± 4.9

13.1 ± 3.2

.018

.017

<.001

Constitutional

symptoms, n (%)

112 (48.9)

11 (52.4)

24 (52.2)

35 (52.2)

70 (57.4)

7 (17.5)

.598

.001

<.001

Allogenic HSCT, n (%)

48 (21.0)

3 (14.3)

10 (21.7)

13 (19.4)

27 (22.1)

8 (20.0)

.800

>.999

.895

Ruxolitinib exposure before HSCT, n (%)

33 (14.4)

2 ( 9.5)

10 (21.7)

12 (17.9)

21 (17.2)

0 ( 0.0%)

>.999

.012

.017

Treatment of non-transplant patients*, n (%)

181 (79.0)

18 (85.7)

36 (78.3)

54 (80.6)

95 (77.9)

32 (80.0)

0.8

>.999

.895

Ruxolitinib

136 (59.6)

17 (81.0)

29 (63.0)

46 (68.7)

82 (67.8)

8 (20.0)

>.999

<.001

<.001

Androgens

47 (20.6)

3 (14.3)

8 (17.4)

11 (16.4)

31 (25.6)

5 (12.5)

0.205

.787

.124

Hydroxyurea

91 (39.9)

14 (66.7)

26 (56.5)

40 (59.7)

31 (25.6)

20 (50.0)

<.001

.437

<.001

Anagrelide

67 (29.4)

4 (19.0)

26 (56.5)

30 (44.8)

14 (11.6)

23 (57.5)

<.001

.283

<.001

Thalidomide

1 (0.4)

0 ( 0.0)

0 ( 0.0)

0 ( 0.0)

1 ( 0.8)

0 ( 0.0)

>.999

-

.641

Investigational agent

13 (5.7)

2 ( 9.5)

2 ( 4.3)

4 ( 6.0)

5 ( 4.1)

4 (10.0)

.835

.699

.379

Leukemic transformation, n (%)

8 (3.5)

0 (0.0)

1 (2.2)

1 (1.5)

6 (4.9)

1 (2.5)

.429

>.999

.838

DIPSS Karyotype, n (%)

*Unfavorable

17 (7.4)

2 (9.5)

4 (8.7)

6 (9.0)

11 (9.0)

0 (0.0)

>.999

.130

.109

MIPSS Karyotype, n (%)

.035

.123

.016

† Favorable

184 (80.3)

18 (85.7)

38 (82.6)

56 (83.5)

90 (73.8)

38 (95.0)

   Intermediate

33 (14.4)

2 (9.5)

3 (6.5)

5 (7.5)

26 (21.3)

2 (5.0)

†† Very High risk

12 (5.2)

1 (4.8)

5 (10.9)

6 (9.0)

6 (4.9)

0 (0.0)

Mutation, n (%)

JAK2V617F

117 (51.1)

21 (100.0)

22 (47.8)

43 (64.2)

54 (44.3)

20 (50.0)

.014

.215

.653

CALR

60 (26.2)

0 (0.0)

18 (39.1)

18 (26.9)

35 (28.7)

7 (17.5)

.922

.383

.233

    Type1/like

39 (17.0)

0 (0.0)

9 (19.6)

9 (13.4)

25 (20.5)

5 (12.5)

.320

.377

.534

    Type2/like

16 (7.0)

0 (0.0)

8 (17.4)

8 (11.9)

7 (5.7)

1 (2.5)

    Others

5 (2.2)

0 (0.0)

1 (2.2)

1 (1.5)

3 (2.5)

1 (2.5)

MPL

10 (4.4)

0 (0.0)

3 (6.5)

3 (4.5)

5 (4.1)

2 (5.0)

>.999

>.999

>.999

ASXL1      

66 (28.8)

3 (14.3)

16 (34.8)

19 (28.4)

44 (36.1)

3 (7.5)

.361

.020

.001

CBL        

5 (2.2)

1 (4.8)

0 (0.0)

1 (1.5)

4 (3.3)

0 (0.0)

.796

>.999

.567

CUX1       

4 (1.7)

0 (0.0)

0 (0.0)

0 (0.0)

4 (3.3)

0 (0.0)

.332

.567

DNMT3A     

11 (4.8)

3 (14.3)

0 (0.0)

3 (4.5)

4 (3.3)

4 (10.0)

.988

.475

.200

EZH2       

5 (2.2)

1 (4.8)

2 (4.3)

3 (4.5)

2 (1.6)

0 (0.0)

.491

.452

>.999

IDH1       

4 (1.7)

1 (4.8)

1 (2.2)

2 (3.0)

2 (1.6)

0 (0.0)

.931

.715

>.999

IDH2       

4 (1.7)

1 (4.8)

0 (0.0)

1 (1.5)

1 (0.8)

2 (5.0)

>.999

.647

.305

NOTCH1     

3 (1.3)

0 (0.0)

1 (2.2)

1 (1.5)

1 (0.8)

1 (2.5)

>.999

>.999

.992

NRAS       

5 (2.2)

0 (0.0)

0 (0.0)

0 (0.0)

3 (2.5)

2 (5.0)

.493

.267

.780

RUNX1      

8 (3.5)

0 (0.0)

3 (6.5)

3 (4.5)

4 (3.3)

1 (2.5)

.988

>.999

>.999

SETBP1     

5 (2.2)

0 (0.0)

0 (0.0)

0 (0.0)

3 (2.5)

2 (5.0)

.493

.267

.780

SF3B1      

17 (7.4)

0 (0.0)

5 (10.9)

5 (7.5)

8 (6.6)

4 (10.0)

>.999

.922

.709

SRSF2      

5 (2.2)

0 (0.0)

0 (0.0)

0 (0.0)

3 (2.5)

2 (5.0)

.493

.267

.780

TET2       

28 (12.2)

4 (19.0)

5 (10.9)

9 (13.4)

12 (9.8)

7 (17.5)

.610

.771

.306

TP53       

8 (3.5)

2 (9.5)

1 (2.2)

3 (4.5)

5 (4.1)

0 (0.0)

>.999

.452

.439

U2AF1      

11 (4.8)

0 (0.0)

1 (2.2)

1 (1.5)

8 (6.6)

2 (5.0)

.227

.647

>.999

U2AF1Q157

3 (1.3)

0 (0.0)

0 (0.0)

0 (0.0)

3 (2.5)

0 (0.0)

.493

.745

ZRSR2      

4 (1.7)

1 (4.8)

1 (2.2)

2 (3.0)

1 (0.8)

1 (2.5)

.595

>.999

.992

Comment 3: in line 155 authors refer to figure 1 but it is missing. lines 184 - 206: there is no figure 1 therefore it is not possible to make any comment on this.

- We apologize for omitting Figure1. It was on the zip file only. I added Figure1 in the manuscripts.

After revision: Figure1 part, below is modified (in blue) (Line 188)

Figure 1. Mutational spectrum in overt primary myelofibrosis (PMF), secondary myelofibrosis (SMF), and prefibrotic myelofibrosis (pre-PMF). (A) Distribution of mutations according to the MIPSS karyotype and RNA splicing genes: SF3B1, SRSF2, ZRSR2, and U2AF1. (B) Proportion of genetic mutations. (C) Proportion of variant allele frequencies (VAF). (D) Chord diagram showing the correlation between clinical variables and mutations. The red bar indicates a positive correlation and the blue bar indicates a negative correlation. The red and blue bars indicate positive and negative correlations, respectively. (E) Decision tree classifying myelofibrosis (MF) by clinical and mutational criteria.

Comment 4: in line 164 authors should refer to Table 1.

- We understand it should be in the comments for clarity. We added mention.

After revision: 3.2. Genetic features of patients with prefibrotic, overt PMF, and secondary MF part, below is modified (in blue) (Line 175)

However, it was found that the proportion of ASXL1 mutations in the pre-PMF group was lower than that in the SMF and PMF groups (Table 1).

Comment 5: Considering karyotype classification (lines 141 - 146) it is not immediately clear what the authors refer to. When they say, in text and table, DIPSS karyotype and MIPSS karyotype they should specify that they are referring to the different strategies adopted through the years for the classification of karyotype abnormalities. would it be possible to state it more clearly in the main text or in table 1 that DIPSS karyotype and MIPSS karyotype are the classification methods adopted by the two scoring systems?

- We clarified the chromosome abnormalities and added them to Table1.

After revision: 3.1. Clinical and cytogenetic features of patients with prefibrotic, overt PMF, and secondary MF part, below is modified (in blue) (Line 158)

* Unfavorable karyotype: Complex karyotype or single or tow abnormalities including +8, -7/7q-, i(17q), -5/5q-, 12p-, inv(3), or 11q23 rearrangements.

† Favorable karyotype normal karyotype, sole abnormalities of 20q-, 13q-, +9, chromosome 1 translocation/duplication, - sex chromosome abnormality including –Y;

††Very High Risk karyotype: single/multiple abnormalities of -7, i(17q), inv(3)/3q21, 12p-/12p11.2, 11q-/11q23, or other autosomal trisomies not including + 8/ + 9 (e.g., +21, +19)

Comment 6: lines 172 - 182: authors highlight that there is a negative correlation between JAK2 CALR and MPL mutations in SMP PMF and pre-PMF patients. This is not a novel result, it has been widely described in literature before. They should not underline this result, they must state that this is not a novel result in line with alrready published literature. JAK2 CALr and MPL mutations are considered as driver mutations since they are mutually exclusive in MPN patients.  In supplementary figure S1 A B C D the gene symbols can't be read therefore the plots can't be understood. Please write gene symbols on the top of columns and on the side of rows. why are there some crosses over dots and numbers in panel B?

- We agree with your concern about too many descriptions which do not interest readers. We rewrote the paragraph. And we revised supplementary figures as per your recommendations.

After revision: 3.2. Genetic features of patients with prefibrotic, overt PMF, and secondary MF part, below is modified (in blue) (Line 181)

Next, we analyzed the correlation of gene mutations detected in more than three patients (Supplementary Figure S1A), and investigated co-occurrence driver mutations in JAK2, CALR, and MPL with other mutations. In the PMF group, mutated JAK2 and TP53 showed we found negative correlations between and mutations in CALR (coefficient of correlation = −0.58, P < 0.001), JAK2, and TP53 (−0.19, P = 0.033). We further found that And CALR and TET2 were positively correlated (0.31, P < 0.001), whereas the MPL mutations were positively correlated with the CBL mutations (0.19, P = 0.032) (Supplementary Figure S1B). In the SMF group, we found that mutated JAK2 was negatively correlated with CALR (−0.81, P < 0.001) and MPL (−0.29, P = 0.017) mutations (Supplementary Figure S1C). In the pre-PMF group, we identified a negative correlation between mutated JAK2 and mutations in CALR (−0.46, P = 0.003) and TET2 (−0.33, P = 0.038) (Supplementary Figure S1D). However, we did not detect any significant correlation between the mutations in CALR and MPL with those in other genes.

Comment 7: in Table 3 it is not clear to me why for many analyses there is no value reported for HR or CI. What does "-" stands for? In my opinion, it is quite surprising that MYSEC-PM is not able to efficently stratify SMF patients in different risk categories since it was designed specifically for this class of patients. Can the authors comment on this?

- “-“stand for the confidence interval is infinite. Patients with pre-PMF were lower mortality in our cohort resulting in no meaningful difference among the group in OS.   We intend to describe OS and PFS together in the Table 3. A similar result showed patients with SMF who had survived for a long time with active intervention. Consequently, MYSEC-PM systems are relatively lesser robust in our study. Our study cohort was limited to a single center, so there is a limitation to be applied generally, which was mentioned in discussion part. Another possible explanation may be due to racial characteristics. Korean people would be lesser exposure to progress-related death such as cerebrovascular accidents or thrombosis.

Comment 8: in line 217 do the authors refer to "overt" myelofibrosis?

- Yes, we understand confusing wording, so we revised the sentence.

After revision 3.4. Distribution of risk categories, outcomes, and prognostic effect of risk stratification systems in each subgroup part, below is modified (in blue) (Line 227)

 In terms of PFS, we found that scores could predict the progression of pre-PMF to either overt-PMF or secondary acute myeloid leukemia (AML) in IPSS, DIPSS DIPSS-plus, MIPSS70, and MYSECPM but not in GIPSS.

Comment 9: I can't understand what the authors mean in line 219 - 220. Can they clarify their conclusion

à We agree that the sentence (line 219-220) have to be clarified. So we rewrote sentence.

After revision 3.4. Distribution of risk categories, outcomes, and prognostic effect of risk stratification systems in each subgroup part, below is modified (in blue) (Line 230)

. We identified that the contribution of clinical variables to the differences between pre-PMF and other overt MF was higher than of genetic factors  These results implied that prognostic difference between pre-PMF and other MF depended on the effect of clinical variables than that of genetic variables.

Comment 10: line 237 - 240: authors say that 48 patients underwent allogenic HSCT. Were these patients considered for survival analysis? If yes, why they were not excluded? In my opinion these patients should have been excluded from the present study, at least for survival analysis, or authors should have consider the date of transplantation as censoring time for these patients since this treatment modality may generate confunding results. Indeed in their pubblication Passamonti et al. (https://doi.org/10.1182/blood-2009-09-245837) adopted the second strategy while Gangat et al (DOI: 10.1200/JCO.2010.32.2446) performed both analyses in order to exclude the introduction of any bias. Can the author perform analysis considering the time of transplantation as censoring time?

- We did subtract patients with HSCT and censored them from HSCT day to show better results. However, a similar trend was observed. For examples, Table in below are showing HSCT censored results.

After revision: 3.4. Distribution of risk categories, outcomes, and prognostic effect of risk stratification systems in each subgroup (in blue) (Line 232)

We identified that the contribution of clinical variables to the differences between pre-PMF and other overt MF was higher than of genetic factors. These results implied that prognostic differences between pre-PMF and other MF depended more on the effect of clinical variables than that of genetic variables. The survival outcomes did not change when patients who underwent transplantation were censored at the time of their transplant.

Variable

Overall survival

*Progression-free survival

HR

95% CI

P

HR

95% CI

P

pre-PMF

Risk stratification

IPSS, INT2 or high

>0.9

17.7

3.04, 103

0.001

DIPSS, INT2 or high

>0.9

18.5

3.22, 107

0.001

DIPSS plus, INT2 or high

>0.9

9.16

1.65, 50.7

0.011

MIPSS70, high

>0.9

9.8

1.90, 50.4

0.006

MIPSS70 +Ver2, high or very high

>0.9

6.04

1.19, 30.6

0.03

GIPSS, INT2 or high

11.4

0.60, 217

0.1

5.13

0.91, 29.0

0.064

MYSECPM, INT2 or high

>0.9

23.9

3.25, 176

0.002

PMF

Risk stratification

IPSS, INT2 or high

3.49

0.92, 13.2

0.066

3.67

0.99, 13.6

0.052

DIPSS, INT2 or high

7.17

1.79, 28.7

0.005

4.84

1.38, 16.9

0.014

DIPSS plus, INT2 or high

6.45

0.83, 50.1

0.075

6.86

0.89, 52.9

0.065

MIPSS70, high

8.36

2.17, 32.1

0.002

5.64

1.68, 19.0

0.005

MIPSS70 +Ver2, high or very high

5.16

1.12, 23.7

0.035

5.77

1.27, 26.2

0.023

GIPSS, INT2 or high

2.26

0.71, 7.17

0.2

2.68

0.87, 8.23

0.086

MYSEC-PM, INT2 or high

3.63

0.92, 14.3

0.066

3.53

1.03, 12.2

0.046

SMF

Risk stratification

IPSS, INT2 or high

1.53

0.13, 17.8

0.7

1.53

0.13, 17.8

0.7

DIPSS, INT2 or high

2.04

0.18, 22.7

0.6

2.04

0.18, 22.7

0.6

DIPSS plus, INT2 or high

0.97

0.08, 11.0

>0.9

0.97

0.08, 11.0

>0.9

MIPSS70, high

5.57

0.50, 62.1

0.2

5.57

0.50, 62.1

0.2

MIPSS70 +Ver2, high or very high

>0.9

>0.9

GIPSS, INT2 or high

>0.9

>0.9

MYSECPM, INT2 or high

3.66

0.33, 41.2

0.3

3.66

0.33, 41.2

0.3

Comment 11: paragraph 3.6: Grinfeld and colleagues in 2018 proposed a genetic classification for patients affected by Philadelphia negative myeloproliferative neoplasms (doi: 10.1056/NEJMoa1716614). Their classification included 6 cathegories based on the presence/absence of specific mutations and and chromosomal abnormalities. Can the authors verify whether classification proposed by Grinfeld and colleagues is effective also in the presented cohort.

- We applied eight subgroups classification by Grinfeld et al. We plotted the figures in the bellow; (A) OS of all patients (B) PFS of patients with PMF (C) PFS of patients with SMF (D) PFS of patients with pre-PMF. The cluster 2 and the MPN with TP53 disruption or aneuploidy group showed similar trends along with cohort. However, when it comes to myelofibrosis subgroups, it is difficult to interpret the results. It might be due to small numbers of cohorts with compared to many subclassifications.

Figure (A) OS for whole cohort(Included in the file)

Figure (B) PFS for PMF (Included in the file)

Figure (C) PFS for SMF(Included in the file)

Figure (D) PFS for pre-PMF(Included in the file)

 Comment 12: In last results' paragraph why did the authors decided to not consider the HMR mutations identified for PMF patients (i.e. ASXL1, EZH2, IDH1/2, SRSF2, U2AF1) that they cyted in intruduction? Can they provide any result about their relevance as high risk mutations not only in overt PMF but also in SMF and pre-PMF? Moreover in dscussion lines 355 - 357 authors say that their results are not completely in line with those reported by Vannucchi et al. Vannucchi et al. reported, in their pubblication that the presence of at least one mutation affecting HMR genes is an independent risk factor for survival in both cohorts included in their pubblication. Furthermore the same mutations were included in the most recent scoring systems confirming their prognostic relevance. Can the authors analyze the impact of the presence of HMR mutations in the studied cohort considering HMR mutations as whole and not independently?.

- Thanks for pointing it out. It seems that our description was not enough. We applied HMR mutations (ASXL1, EZH2, IDH1/2, SRSF2) which Vannucchi et al. reported in each MF subgroup by MIPSS70, which showed their prognostic relevance in PMF, but not in SMF and pre-PMF. Furthermore, we analyzed as per your request, HMR mutations with U2AF1(ASXL1, EZH2, IDH1/2, SRSF2, U2AF1) below the table. In addition to previous HMR mutations, we wanted to find additional genetic information predicting PFS, that is a leukemic transformation and a marrow fibrosis progression as the myeloproliferative disease goes by, which can applied in pre-MF. For the purpose, we performed univariate analyses in each genetic factor and literature review. And, we applied various genetic combinations.

MF subtypes

5-year PFS in

HMR mutations with U2AF1 vs wild type

p-value

Whole cohorts

60.7 vs 80.1

0.052

PMF

57.3 vs 80.5

0.19

SMF

74.4 vs 81.5

0.6

pre-PMF

42.9 vs 85.3

0.036

Comment 13: In figure 4 authors show the negative impact of wild type SF2B1 or mutations in SRSF2, RUNX1, U2AF1, ASXL1, and TP53 on survival and progression in the study cohort. Did they perform a multivariate analysis? do this "high molecular risk" profile represent a risk factor for survival independent from current prognostic systems?

- Yes, we tried to find independent aspects after adjusting the impact of previous systems on OS and PFS. In the table below, multivariate analysis was performed with each prognostic system.

Multivariate analysis in OS

HR

95% CI

p

IPSS, INT2 or high

6.25

2.09, 18.7

0.001

High molecular risk group

6.09

2.63, 14.1

<0.001

DIPSS, INT2 or high

8.37

2.78, 25.1

<0.001

High molecular risk group

5.31

2.29, 12.3

<0.001

DIPSS plus, INT2 or high

6.92

1.62, 29.6

0.009

High molecular risk group

6.47

2.78, 15.0

<0.001

MIPSS70, high

6.15

2.53, 15.0

<0.001

High molecular risk group

5.19

2.18, 12.3

<0.001

MIPSS70 +Ver2, high or very high

11.7

2.73, 50.3

<0.001

High molecular risk group

5.02

2.16, 11.7

<0.001

GIPSS, INT2 or high

2.05

0.91, 4.61

0.082

High molecular risk group

6.70

2.71, 16.5

<0.001

MYSECPM, INT2 or high

2.28

0.97, 5.38

0.060

High molecular risk group

6.61

2.68, 16.3

<0.001

Multivariate analysis in PFS

HR

95% CI

p

IPSS, INT2 or high

4.12

1.75, 9.70

0.001

High molecular risk group

4.25

2.02, 8.93

<0.001

DIPSS, INT2 or high

4.65

2.04, 10.6

<0.001

High molecular risk group

3.85

1.83, 8.13

<0.001

DIPSS plus, INT2 or high

3.41

1.30, 8.92

0.012

High molecular risk group

4.59

2.18, 9.68

<0.001

MIPSS70, high

3.41

1.67, 6.97

<0.001

High molecular risk group

3.86

1.79, 8.30

<0.001

MIPSS70 +Ver2, high or very high

4.73

1.92, 11.6

<0.001

High molecular risk group

3.73

1.76, 7.89

<0.001

GIPSS, INT2 or high

2.13

1.05, 4.29

0.035

High molecular risk group

4.38

2.00, 9.60

<0.001

MYSECPM, INT2 or high

1.93

0.92, 4.05

0.081

High molecular risk group

4.57

2.06, 10.1

<0.001

Comment 14:

 - line 63 "myeloproliferations" should be "myeloproliferation"

- line 136: why PPV appears before pre-PMF? The number of pre-PMF cases is greater then those of PPV-MF.

- line 209: remove "T"

- lines 236 -237: substitute "With" with "within".

- line 360 correct "Grinfeld"

- Thanks for pointing it out. We corrected them.

After revision:  below is modified (in blue)

(Line 63) Excessive myeloproliferation in Philadelphia-negative MPNs are driven by mutations in JAK2, CALR, MPL, and uncommon variants

(Line 141) Among the patients, PMF was the most common (n = 122, 53.3%), followed by PET-MF (n = 46, 20.1%), pre-PMF (n = 40, 17.4%), and PPV-MF (n = 21, 9.2%).

(Line 220)T The distribution of the risk stratification system in each subgroup is described in Table 2.

(Line 249) We recorded 30 deaths among all patients (22 within PMF, 6 within SMF, and 2 within pre-PMF).

(Line 365) However, Grinfeld et al. reported that leukemic transformation is related to the occurrence of mutations in RUNX1, TP53, and U2AF1 in MPN disease, which are consistent with our results

Round 2

Reviewer 1 Report

No further comments